# DiffuPac: Contextual Mimicry in Adversarial Packets Generation via Diffusion Model

**Abdullah Bin Jasni**
Graduate School of Engineering
Nagaoka University of Technology
Nagaoka, Japan
s203108@stn.nagaokaut.ac.jp

**Akiko Manada**
Graduate School of Engineering
Nagaoka University of Technology
Nagaoka, Japan
amanada@vos.nagaokaut.ac.jp

**Kohei Watabe**
Graduate School of Science and Engineering
Saitama University
Saitama, Japan
kwatabe@mail.saitama-u.ac.jp

## Abstract

In domains of cybersecurity, recent advancements in Machine Learning (ML) and Deep Learning (DL) have significantly enhanced Network Intrusion Detection Systems (NIDS), improving the effectiveness of cybersecurity operations. However, attackers have also leveraged ML/DL to develop sophisticated models that generate adversarial packets capable of evading NIDS detection. Consequently, defenders must study and analyze these models to prepare for the evasion attacks that exploit NIDS detection mechanisms. Unfortunately, conventional generation models often rely on unrealistic assumptions about attackers' knowledge of NIDS components, making them impractical for real-world scenarios. To address this issue, we present DiffuPac, a first-of-its-kind generation model designed to generate adversarial packets that evade detection without relying on specific NIDS components. DiffuPac integrates a pre-trained Bidirectional Encoder Representations from Transformers (BERT) with diffusion model, which, through its capability for conditional denoising and classifier-free guidance, effectively addresses the real-world constraint of limited attacker knowledge. By concatenating malicious packets with contextually relevant normal packets and applying targeted noising only to the malicious packets, DiffuPac seamlessly blends adversarial packets into genuine network traffic. Through evaluations on real-world datasets, we demonstrate that DiffuPac achieves strong evasion capabilities against sophisticated NIDS, outperforming conventional methods by an average of 6.69 percentage points, while preserving the functionality and practicality of the generated adversarial packets.

## 1 Introduction

Network Intrusion Detection Systems (NIDS) play a pivotal role in safeguarding the vast array of digital devices and infrastructures that permeate our lives. As of 2023, the global count of active IoT devices is expected to reach approximately 15.14 billion, with projections suggesting a rise to 30 billion by 2030 (Statista [2023]). This explosive growth, fueled by applications spanning from consumer electronics to industrial automation and healthcare, presents formidable security challenges. To meet these challenges, advancements in Machine Learning (ML) and Deep Learning (DL) have significantly bolstered the efficacy of NIDS in monitoring IoT traffic and detecting malicious activities (Talaei Khoei and Kaabouch [2023]; Talaei Khoei et al. [2023]).

38th Conference on Neural Information Processing Systems (NeurIPS 2024).

However, the rapid evolution of generative AI technologies has ushered in a new era of cybersecurity threats, notably through the creation of adversarial packets designed to evade detection by even the most sophisticated NIDS. These AI-driven attacks can emulate and synthesize legitimate network behaviors, presenting an unprecedented challenge to existing security paradigms. Generative models, particularly those trained on extensive datasets of genuine network traffic, can generate adversarial packets that blend malicious functionalities within seemingly normal packet sequences, thus effectively camouflaging their malicious intents.

In response to these evolving threats, it is critical for cybersecurity defenders to deepen their understanding of these generative AI models. By scrutinizing the mechanisms through which these models generate adversarial packets, defenders can better anticipate and counteract adversarial tactics that compromise the detection capabilities of NIDS (Ibitoye et al. [2019]; Khazane et al. [2024]). The urgency to develop innovative solutions that can adapt to and preempt these adversarial tactics is paramount, ensuring the reliability and robustness of NIDS in an increasingly complex threat landscape.

Traditional methods for generating adversarial packets have primarily relied on direct engagement with NIDS or the use of surrogate classifiers, often assuming unrealistic levels of attacker access to NIDS configurations. Table 1 summarizes recent literature on adversarial packet generation, highlighting the limitations of these approaches. Studies using techniques such as Network Emulator (NetEM) and Metasploit (Homoliak et al. [2018]), Generative Adversarial Network (GAN) and Particle Swarm Optimization (PSO) (Han et al. [2021]), and Reinforcement Learning (RL) (Hore et al. [2023]) demonstrate that while these methods successfully modified the packets behavior, their

Table 1: Summary of recent literature on adversarial packets generation.

| Author | Year | Data Set | Classifier-Techniques | Algorithm |
|---|---|---|---|---|
| Homoliak et al. [2018] | 2018 | ASNM-NBPO | Surrogate Classifier | NetEM, Metasploit |
| Hashemi et al. [2019] | 2019 | CICIDS-2018 | Surrogate Classifier | Trial and error |
| Kuppa et al. [2019] | 2019 | CICIDS-2018 | Surrogate Classifier | Manifold Approx. |
| Han et al. [2021] | 2021 | Kitsune, CICIDS-2017 | NIDS feature extractor | GAN and PSO |
| Sharon et al. [2021] | 2021 | Kitsune, CICIDS-2017 | NIDS classifier | LSTM |
| Hore et al. [2023] | 2023 | CICIDS-2017, CICIDS-2018 | Surrogate Classifier | RL |

efficacy in evading detection in real-world conditions remains suboptimal. The reliance on detailed knowledge of NIDS models is flawed, as attackers typically operate with limited information about the underlying security infrastructure. This gap underscores the necessity for more practical and effective adversarial generation methods that align with the realistic constraints faced by attackers.

To address these challenges, we introduce *DiffuPac: Contextual Mimicry in Adversarial Packet Generation via Diffusion Model*, a novel solution that leverages the combined strengths of Bidirectional Encoder Representations from Transformers (BERT) and diffusion model. This innovative fusion not only promises high accuracy in generating adversarial packets but also operates under the realistic assumption that attackers lack direct access to NIDS models. DiffuPac leverages the extensive contextual understanding provided by BERT, which has been trained on diverse datasets representing a wide range of network behaviors, along with the generative capabilities of diffusion models. This fusion results in a sophisticated adversarial tactics where the elements of the attack are seamlessly integrated into the network traffic, making them indistinguishable from legitimate data. This capability represents a significant leap forward, offering a stealthy approach that outmaneuvers current NIDS through advanced mimicry rather than direct confrontation.

In summary, the principal contributions of this paper are as follows: (a) we have pioneered the integration of BERT and diffusion models to create DiffuPac, marking a first in the cybersecurity domain. This novel methodology sets a precedent in the field by blending the advanced contextual comprehension of network traffic with sophisticated generative capabilities to produce adversarial packets that are both stealthy and indistinguishable from genuine traffic; (b) we introduce a unique concatenation strategy coupled with targeted noising techniques. These innovations ensure that the adversarial packets not only blend seamlessly into the network environment but also dynamically adapt to evade modern detection systems; (c) DiffuPac advances a classifier-free approach to adversarial packet generation. This approach challenges traditional dependency on surrogate classifiers, offering

a new paradigm that more accurately reflects the constraints and capabilities of real-world attackers; (d) Lastly, our extensive experimental evaluations, conducted on real-world datasets, demonstrate that DiffuPac significantly outperforms existing methods in terms of evasion effectiveness, establishing new benchmarks for the generation of adversarial packets. These contributions collectively push the boundaries of what is possible in the realm of network security, paving the way for more resilient cybersecurity defenses and a deeper understanding of adversarial tactics in network environments.

## 2 Related Works

**Adversarial Attacks on ML/DL-based NIDS**. NIDS are crucial for protecting digital infrastructures by monitoring network traffic and identifying potential threats through signature-based and anomaly-based detection paradigms. Signature-based NIDS use pattern matching against predefined threat databases, while anomaly-based NIDS employ machine learning models to detect deviations from benign traffic patterns, providing an advantage in detecting sophisticated threats. The transition to DL in anomaly-based NIDS, as highlighted by Ahmad et al. [2021], has enhanced threat detection due to the capability to learn abstract patterns, though this shift has introduced a vulnerability to adversarial attacks. These attacks modify network data subtly to evade NIDS, a phenomenon first noted in computer vision (Szegedy et al. [2014]) and now challenging DL-based NIDS. Adversarial attack tactics vary based on attacker knowledge, ranging from white-box attacks with full system knowledge to black-box attacks with no classifier knowledge, as outlined by McCarthy et al. [2022]. Black-box scenarios are common in real-world threats since attackers generally lack direct NIDS access, necessitating sophisticated modifications of data to exploit DL model vulnerabilities and blur the distinction between normal and malicious traffic.

**Adversarial Attacks Evading NIDS**. In the study of NIDS, researchers explore feature-level and packet-level attacks to enhance robustness. Feature-level attacks modify input network features using methods like GANs to mislead classifiers without direct knowledge of their mechanisms. For instance, Yang et al. [2018] used transfer-based and score-based attacks to deceive a Deep Neural Network (DNN) model on the NSL-KDD dataset, while Sheatsley et al. [2022] developed an Augmented JSMA (AJSMA) to ensure realistic feature modifications. However, these still fail to generate executable malicious packets since they do not provide a method for converting modified features into packet sequences. Conversely, packet-level attacks, which modify network packets directly to maintain their malicious intent while evading detection, are particularly effective. As detailed in Table 1, studies such as Hashemi et al. [2019] show successful modifications using non-payload based and mimicking operations. In the survey He et al. [2023], authors emphasized that the practicality and replayability of packet-level attacks make them more dominant, ensuring adversarial packets evade detection while remaining executable. Given this efficacy, our research model, DiffuPac, is designed to excel in this domain, outperforming previous models in robustness and detection evasion, and setting a new standard for NIDS efficacy in adversarial cybersecurity.

**Diffusion Models**. Represent a breakthrough in generative modeling, offering a novel framework for understanding data as a dynamic and stochastic process. Based on the works of Sohl-Dickstein et al. [2015] and Ho et al. [2020], these models conceptualize data points $\mathbf{z}_0 \in \mathbb{R}^d$ (where $d$ is a positive integer) as the end result of a reverse Markov chain. In the forward diffusion process, data points $\mathbf{z}_0$ are gradually transformed into a noise distribution . This transformation is modeled by a Markov chain starting from the data distribution $q(\mathbf{z}_0)$ and ending in a Gaussian distribution $\mathbf{z}_T \sim \mathcal{N}(\mathbf{0}, \mathbf{I})$, where $T$ represents the total number of timesteps in the forward diffusion process. The transitions are defined $q(\mathbf{z}_t|\mathbf{z}_{t-1}) = \mathcal{N}(\mathbf{z}_t; \sqrt{1 - \beta_t}\mathbf{z}_{t-1}, \beta_t\mathbf{I})$ for $1 \leq t \leq T$, where $\beta_t$ as the variance scale. In the reverse process, the model $f_\theta$ (usually a U-Net or a Transformer) learns to reconstruct the original data $\mathbf{z}_0$ from the noised data $\mathbf{z}_T$. It does so by iteratively estimating the parameters $p_\theta(\mathbf{z}_{t-1}|\mathbf{z}_t) = \mathcal{N}(\mathbf{z}_{t-1}; \mu_\theta(\mathbf{z}_t, t), \sigma_t^2\mathbf{I})$, where $\mu_\theta$ and $\sigma_t^2$ predict the mean and variance of the distribution.

**BERT and Diffusion Models Approaches in Cybersecurity and Other Fields**. The application of pre-trained models, such as ET-BERT (Lin et al. [2022]) and NetGPT (Meng et al. [2023]), has revolutionized understanding of network traffic by capturing the intricacies of language and interpreting complex network patterns. Meanwhile, diffusion models have emerged as powerful generative tools capable of producing high-fidelity data for applications like image generation and notably, synthetic network traffic generation. These models leverage a unique denoising training objective to closely mimic the original data distribution, making them promising for generating realistic network

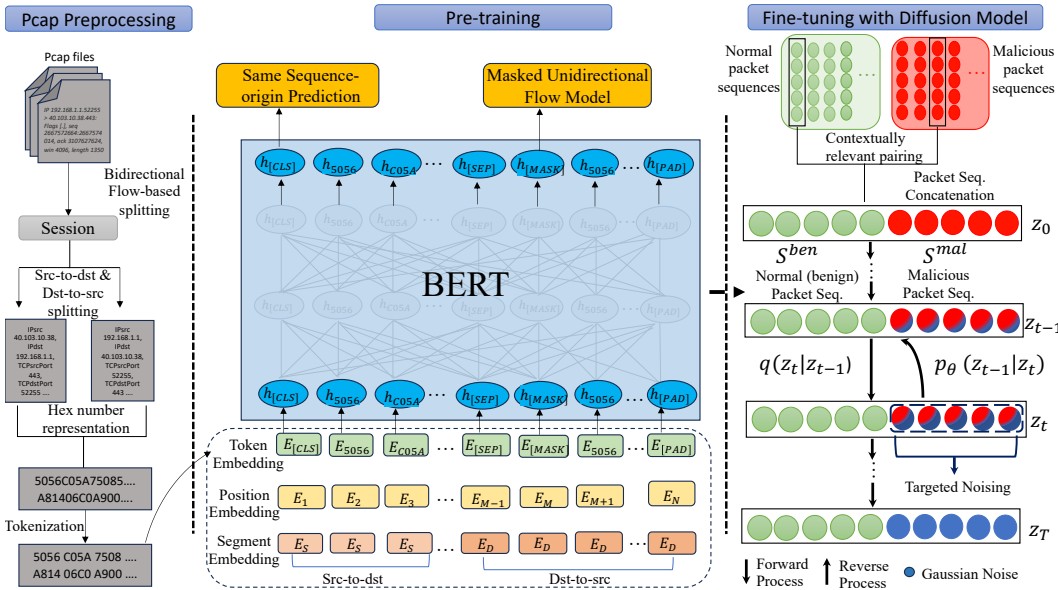

Figure 1: Proposed arhitecture is divided into three phases: pcap pre-processing, pre-training and fine-tuning with diffusion models.

traffic for testing and analysis. Despite these advancements, combining the contextual understanding of pre-trained models with the generative prowess of diffusion models remains underexplored in adversarial packet generation. This integration offers a synergistic approach, merging the nuanced comprehension of network behaviors by pre-trained models with the high-quality data generation of diffusion models. By leveraging these technologies together, cybersecurity researchers can enhance the realism of simulated network environments and develop more advanced evasion tactics.

## 3 DiffuPac

In this section, we introduce DiffuPac, a first-of-its kind adversarial packet generation model. Figure 1 illustrates the overall framework of the proposed model, consisting of three main phases: pcap pre-processing, pre-training and fine-tuning with diffusion model.

### 3.1 Data pre-processing

In real network environments, traffic contains diverse flows from various applications, protocols, and services, complicating the learning of stable representations. Therefore, we first split pcap (packet capture) files into sessions (bidirectional flows) based on IP addresses, port addresses, and protocols. To refine the training of the BERT model, we further split sessions into unidirectional flows, categorizing them as either source-to-destination (src-to-dst) packet sequence or destination-to-source (dst-to-src) packet sequence. This categorization is crucial for the BERT model's pre-training. Network traffic varies due to diverse protocols and network services, resulting in different formats and patterns. To handle these variations and encoding requirements, we convert each byte to its corresponding hex number and tokenize using WordPiece (Wu et al. [2016]). Each token ranges from 0 to 65535, with a dictionary size of 65536. We also incorporate special tokens [CLS], [SEP], [PAD], and [MASK] for training tasks. [CLS] is used at the beginning of each sequence and helps in classification tasks. [SEP] separates different sequences or segments within the same input. [PAD] ensures sequences are of uniform length and satisfy the minimum length requirement. [MASK] is used in pre-training task, where it temporarily substitutes tokens to be predicted.

### 3.2 Pre-training

As shown in Figure 1, the input tokens are processed using an embedding strategy that involves the sum of three types of embeddings: token embeddings, positional embeddings, and segment embeddings. Each embedding has a dimension of 768. *Token Embeddings* are high-dimensional

vectors that uniquely represent each token, acting as their exclusive identifiers. *Positional Embeddings* are used to capture the temporal relationships of tokens, ensuring that the model learns to focus on the order of data transmission. *Segment Embeddings* differentiate packets within a single flow, as packets may not inherently share semantic associations, and preserving the order of packets to maintain the temporal sequence of events in a session flow.

We design our pre-training tasks based on the approach described in Devlin et al. [2019]. Our two proposed pre-training tasks aim to capture the contextual relationships between traffic bytes. The first task involves predicting masked tokens to learn the underlying patterns and dependencies within the traffic data. The second task predicts the transmission order by determining whether packets belong to src-to-dst or dst-to-src sequences, thereby capturing the directional flow of the network traffic.

**Masked Unidirectional Flow Model**. Inspired by BERT's Masked Language Model (MLM) in natural language processing, we adapt this approach for network traffic analysis through our Masked Unidirectional Flow Model. This model is designed to understand and predict the semantic patterns within bidirectional network flows; that are src-to-dst sequence and dst-to-src sequence. During pre-training, each token in the input sequence is masked with a probability of 15%. Amongst these masked tokens 80% are replaced with a [MASK] token, 10% are replaced with a random token from the vocabulary, and 10% are left unchanged. This introduces variability that mimics real-world data inconsistencies. The training objective is to minimize the negative log-likelihood of correctly predicting the original tokens at the masked positions. Formally, the loss function $L_{\mathrm{MUM}}$ for this task is defined as:

$$L_{\mathrm{MUM}} = -\sum_{i=1}^{n} 1(m_i) \log P(v_i \mid V_{\mathrm{masked}}; \theta), \tag{1}$$

where $n$ is the total number of tokens in the sequence, $1(m_i)$ is an indicator function that is 1 if the token $v_i$ was masked (0 otherwise). $m_i$ indicates the masking status of the $i$-th token and $V_{\mathrm{masked}}$ is the masked sequence. $v_i$ is the actual token at position $i$ and $P(v_i \mid V_{\mathrm{masked}}; \theta)$ is the probability of predicting the original token $v_i$ for given masked sequence and model parameters $\theta$. The transformer encoder, characteristic of BERT, processes $V_{\mathrm{masked}}$ to predict each masked token. This architecture leverages self-attention mechanisms to capture dependencies and context effectively, which is crucial for understanding complex patterns in network traffic.

**Same Sequence-origin Prediction**. Inspired by Lin et al. [2022], this task employs a dedicated binary classifier to determine the directional origin of network traffic, specifically whether packets in a sequence originate from src-to-dst or dst-to-src. This classification enhances the model's understanding of network flows and the contextual relationships between packets, which is crucial for recognizing communication patterns and dependencies in network traffic. For this task, each packet $a_r$ is paired with another packet $a_s$. 50% of the time, $a_s$ is the next logical packet in the flow (src-to-dst or dst-to-src); the other 50% of the time, it is a randomly chosen packet from the opposite flow. The classifier then predicts whether $a_s$ follows $a_r$ in the correct directional sequence. This pairing strategy improves the model's capability to discern patterns in packet flows. Let $A = \{(a_r, a_s)\}$ be the set of packet pairs, where each pair is labeled with $b_w \in \{0, 1\}$ (1 if $a_s$ follows $a_r$ in the correct flow, 0 otherwise). The loss function $L_{\mathrm{SSP}}$ for this task can be formulated as:

$$L_{\mathrm{SSP}} = -\sum_{w=1}^{K} b_w \log P(b_w = 1 \mid a_r, a_s; \theta) + (1 - b_w) \log(1 - P(b_w = 1 \mid a_r, a_s; \theta)), \tag{2}$$

where $K$ is the total number of packet pairs and $\theta$ represents the trainable parameters of the classifier. $P(b_w = 1 \mid a_r, a_s; \theta)$ is the probability predicted by the classifier that $a_s$ correctly follows $a_r$ in the given network flow direction.

In summary, the final pre-training objective is the sum of the above two losses, which can defined as:

$$L = L_{\mathrm{SSP}} + L_{\mathrm{MUM}}. \tag{3}$$

Fu et al. [2021] mentioned that the number of packets within each flow can vary significantly. Given the constraints imposed by packet size and the potential volume of network traffic, computational efficiency is a paramount concern. In addressing this problems, as demonstrated in Dai et al. [2023] that the initial packets in a flow contain the most significant information, we limit our analysis to the first three packets of each heavy flow. This means that for each session, we analyze a total of six packets—three from the src-to-dst flow and three from the dst-to-src flow. This strategy ensures a comprehensive view of the session while maintaining efficiency.

### 3.3 Fine-tuning with Diffusion Models

**Forward Process with Packet Sequences Concatenation Strategy**. In the fine-tuning phase, the goal is to train the model so that malicious packets can mimic normal packets to bypass NIDS. This involves using packet sequences that are only from the src-to-dst, reflecting the adversary's control over the packets being sent (Hore et al. [2023]). The forward process of our diffusion model begins by embedding both normal (benign) packet sequences $\mathbf{S}^{\mathrm{ben}}$ and malicious packet sequences $\mathbf{S}^{\mathrm{mal}}$. This transformation of discrete packet data into a continuous feature spaces uses an embedding function adapted from Li et al. [2022]. Building upon the groundwork of DiffSeq (Gong et al. [2023]) approach, which typically involves random merging, our model innovates by leveraging the deep contextual insights provided by the pre-trained BERT model (Details in A.2). This allows us to strategically pair normal and malicious sequences with strong contextual alignments that show similarity in network behavior patterns. These contextually aligned pairs are crucial for mimicking normal traffic and increasing the chances of bypassing NIDS. By integrating these contextually aligned pairs into our diffusion process, the model extends the original forward chain to a new Markov transition $q_\phi(\mathbf{z}_0 \mid \mathbf{S}^{\mathrm{ben \oplus mal}}) = \mathcal{N}(\mathrm{EMB}(\mathbf{S}^{\mathrm{ben \oplus mal}}), \beta_0 \mathbf{I})$, where $\mathrm{EMB}(\mathbf{S}^{\mathrm{ben \oplus mal}})$ symbolizes the embedding transformation and concatenation of normal and malicious packet sequences. While $\mathbf{S}^{\mathrm{ben \oplus mal}}$ denotes the initial concatenated sequence, the forward process gradually perturbs this initial state $\mathbf{z}_0$ through a series of transitions, producing latent variables $\mathbf{z}_1, \mathbf{z}_2, \ldots, \mathbf{z}_t$.

**Targeted Noising**. To enhance our diffusion model's capability for adversarial packet generation, we simplify the model's state transitions by defining $\mathbf{z}_t = \mathbf{x}_t \oplus \mathbf{y}_t$, where $\mathbf{x}_t$ and $\mathbf{y}_t$ correspond to the portions of $\mathbf{z}_t$ that belong to $\mathbf{S}^{\mathrm{ben}}$ and $\mathbf{S}^{\mathrm{mal}}$, respectively. This setup allows us to strategically inject noise into only the malicious packet sequences (represented by $\mathbf{y}_t$), rather than the entire state $\mathbf{z}_t$, during each forward step $q(\mathbf{z}_t \mid \mathbf{z}_{t-1})$. This targeted noising, inspired by DiffSeq, is pivotal for adapting conventional diffusion models for targeted modification.

**Reverse Process With Normal Packet Guidance**. Building upon the foundational principles of DiffSeq, our model introduces an innovative reverse process tailored for more nuanced handling of network traffic data. This process distinctively utilizes normal packet sequences as a guiding framework, enabling a sophisticated denoising technique that treats the concatenated sequences of normal and noise-added malicious packets as a unified unit. This approach effectively "teaches" the model to perceive these malicious elements as integral parts of the normal traffic pattern. A key aspect of this reverse process is the use of the pre-trained BERT model as the denoising engine during packet reconstruction. In this phase, the BERT model undergoes fine-tuning within the diffusion framework, ensuring that it is specifically adapted for reconstructing packets. As a result, the BERT model is not only responsible for contextual understanding but also plays a central role in recovering malicious packets that have been integrated into normal traffic patterns. The conditional denoising process effectively employs Bayesian inference to parameterize the transition probabilities between states, ensuring precise control over each step in the reverse process. These parameterizations are mathematically articulated through the equations: $p_\theta(\mathbf{z}_{0:T}) := p(\mathbf{z}_T) \prod_{t=1}^{T} p_\theta(\mathbf{z}_{t-1}|\mathbf{z}_t)$ and $p_\theta(\mathbf{z}_{t-1}|\mathbf{z}_t) = \mathcal{N}(\mathbf{z}_{t-1}; \mu_\theta(\mathbf{z}_t, t), \sigma_\theta(\mathbf{z}_t, t))$, where $\mu_\theta(\cdot)$ and $\sigma_\theta(\cdot)$ model the mean and variance necessary for the stochastic reverse transition from $\mathbf{z}_t$ to $\mathbf{z}_{t-1}$, respectively. To rigorously ensure that the malicious packets are not only recovered but also convincingly mimic normal packets, our model optimizes a specially formulated variational lower bound $L_{VLB}$. This objective underscores the model's effectiveness in blending malicious packets within normal packets:

$$\min_\theta L_{VLB} = \min_\theta \left[ \sum_{t=2}^{T} \|\mathbf{z}_0 - f_\theta(\mathbf{z}_t, t)\|^2 + \|\mathrm{EMB}(\mathbf{S}^{\mathrm{ben \oplus mal}}) - f_\theta(\mathbf{z}_1, 1)\|^2 - \log p_\theta(\mathbf{S}^{\mathrm{ben \oplus mal}} \mid \mathbf{z}_0) \right]$$

(4)

This objective is particularly focused on accurately reconstructing the initial state $\mathbf{z}_0$ from the noised states, with a distinct emphasis on ensuring that the malicious components are seamlessly integrated into the normal packets pattern.

**Preserving Packets' Integrity**. We propose an innovative approach that utilizes a parallel data structure called *"model dictionary"* that clearly distinguishes between mutable and immutable fields within each packet. By categorizing fields into mutable (e.g. TCP flags, TTL, and window size) and immutable (including the critical 5-tuple information and payload), we ensure that modifications during training do not compromise the packet's integrity. The model dictionary serves a dual purpose: it retains the original values of immutable fields and establishes a link to their mutable counterparts. This approach allows for dynamic modification of mutable fields during training without affecting

the core attributes of the packet. After training, any changes made to mutable fields are seamlessly integrated with the preserved immutable fields. This recombination ensures that the modified packets maintain their operational integrity and are indistinguishable from genuine traffic in real-world scenarios. By bypassing traditional domain constraints—which typically complicate training and pose convergence challenges—our method simplifies the training process and obviates the need for additional compensatory loss functions (Letcher [2021]). The outcome is a highly effective generation of adversarial packets that are capable of evading detection without compromising the essential characteristics of the original packets.

# 4 Experiments

We conduct experiments to validate the performance of DiffuPac on 6 types of attacks, against 6 classifiers.

## 4.1 Experimental Setup

**Dataset**. The datasets used for this model are Kitsune Dataset (Mirsky et al. [2018]) and CICIDS-2017 Dataset (Sharafaldin et al. [2018]). Initially, *pre-training of the BERT model* utilizes a large subset of unlabeled network traffic to leverage the model's capability to capture diverse traffic patterns, accounting for 60% of the total data. *Fine-tuning phase*, the focus shifts to a smaller, labeled dataset, which constitutes 20% of the total data. This dataset is distinctly partitioned into malicious and normal packets, which is crucial for training the model to mimic malicious packets as normal. *Training the classifier and NIDS* then utilizes another 10% of the total data consisting of labeled portion. *Testing phase* is conducted with the remaining 10% of the data, reserved exclusively for evaluating the model's efficacy. This phase includes testing both original and mimicked malicious packets to rigorously assess the model's real-world applicability and its capability to generalize across unseen data.

**Baseline Models**. We evaluate two baseline models—Traffic Manipulator (Han et al. [2021]) and TANTRA (Sharon et al. [2021])—both of which represent current strategies for evading NIDS but share a significant limitation in their reliance on specific operational assumptions. Traffic Manipulator formulates the evasions as a bi-level optimization problem, where the first level uses GAN to identify adversarial features closely mimicking the original network features, and the second level utilizes PSO to modify packet characteristics to exploit vulnerabilities in NIDS. TANTRA employs a three-step process involving an LSTM trained to model benign traffic by predicting inter-packet delays. This LSTM then adjusts malicious traffic to mimic these benign patterns before deployment. Both models, however, assume a degree of accessibility to NIDS configurations or feature extractors that is often unrealistic in practical scenarios. This reliance on specific knowledge about NIDS configurations creates a critical dependency that can limit the deployment and scalability of these evasion techniques in diverse operational environments. Conversely, DiffuPac operates under the assumption of zero prior knowledge about NIDS configurations or feature extractors.

**Implementation Details**. DiffuPac leverages a BERT model with 12 transformer blocks, each featuring 12 attention heads and an embedding dimension of 768, to capture nuanced relationships in network traffic. This setup supports the maximum sequence length of 512. The diffusion model incorporates time step embedding to provide temporal context, enhancing the model's understanding of the reverse process. To manage the computational demands, particularly during the sampling phase, a projection layer reduces the embedding dimension to 128. Additionally, DiffuPac utilizes the FAISS library (Douze et al. [2024]), renowned for its capability to handle large-scale similarity search and clustering of dense vectors.

**Experimental Details**. Our experimental framework integrates advanced feature extraction tools and a diverse set of machine learning classifiers, building on methodologies established in the Traffic Manipulator research. We employ 2 feature extractors: AfterImage (Mirsky et al. [2018]), which provides detailed packet statistics, and CICFlowMeter (Draper-Gil et al. [2016]), which assesses connection-level metrics. Recognizing the documented shortcomings of the original CICFlowMeter, our research employs a revised version that integrates enhancements and corrections proposed in recent studies (Engelen et al. [2021], Liu et al. [2022]). These modifications are crucial in ensuring more accurate and reliable feature extraction for our analysis. AfterImage and CICFlowMeter offer a comprehensive view of network traffic, capturing both packet-level and flow-level data essential

Table 2: Comparative analysis of attack detection and evasion rates.

(a) Botnet

| Feature Extractor | Classifier | Detection | | | Evasion Rate (*MER*) | | |
|---|---|---|---|---|---|---|---|
| | | P | R | F1 | GAN & PSO | LSTM | Ours |
| CIC FLowMeter | KitNET | 0.84 | 0.94 | 0.92 | 37.24% | **50.69%** | 46.48% |
| | DT | 0.79 | 0.91 | 0.82 | 35.88% | 49.62% | **60.98%** |
| | IF | 0.99 | 0.90 | 0.95 | 39.78% | 42.93% | **49.71%** |
| | MLP | 0.92 | 0.84 | 0.86 | 41.05% | 53.87% | **63.05%** |
| | SVM | 0.99 | 0.92 | 0.95 | 88.79% | **91.95%** | 64.49% |
| | LR | 0.84 | 0.91 | 0.89 | 24.72% | 30.52% | **42.08%** |
| AfterImage | KitNET | 0.96 | 0.90 | 0.94 | 99.18% | **99.79%** | 74.46% |
| | DT | 0.79 | 0.90 | 0.84 | 63.42% | 67.30% | **72.13%** |
| | IF | 0.99 | 0.90 | 0.94 | 31.48% | **61.08%** | 52.79% |
| | MLP | 0.96 | 0.97 | 0.97 | 48.60% | 60.97% | **64.92%** |
| | SVM | 0.99 | 0.90 | 0.94 | 40.31% | 51.24% | **69.19%** |
| | LR | 0.96 | 0.90 | 0.93 | 53.28% | 50.70% | **58.98%** |

(b) MITM

| Feature Extractor | Classifier | Detection | | | Evasion Rate (*MER*) | | |
|---|---|---|---|---|---|---|---|
| | | P | R | F1 | GAN & PSO | LSTM | Ours |
| CIC FLowMeter | KitNET | 0.92 | 0.94 | 0.91 | 38.22% | 44.39% | **53.87%** |
| | DT | 0.74 | 0.79 | 0.76 | 49.98% | 57.77% | **64.12%** |
| | IF | 0.99 | 0.92 | 0.94 | 26.74% | 38.64% | **52.99%** |
| | MLP | 0.77 | 0.72 | 0.74 | 52.07% | 43.19% | **73.21%** |
| | SVM | 0.74 | 0.79 | 0.78 | 42.11% | 45.54% | **60.46%** |
| | LR | 0.73 | 0.78 | 0.72 | 35.87% | **50.04%** | 47.48% |
| AfterImage | KitNET | 0.94 | 0.96 | 0.93 | 68.79% | **79.29%** | 58.48% |
| | DT | 0.75 | 0.89 | 0.84 | 53.18% | 58.15% | **70.04%** |
| | IF | 0.81 | 0.83 | 0.86 | 26.53% | 31.27% | **45.71%** |
| | MLP | 0.92 | 0.90 | 0.93 | 50.65% | 59.30% | **71.45%** |
| | SVM | 0.99 | 0.90 | 0.94 | 63.51% | 57.51% | **66.53%** |
| | LR | 0.91 | 0.94 | 0.90 | 44.68% | 46.39% | **52.38%** |

for nuanced analysis. These feature extractors are integral to NIDS detection as they preprocess network data into structured features that are then analyzed by ML classifiers. This preprocessing step is crucial for transforming raw network traffic into a format that classifiers can effectively interpret, thereby enhancing the accuracy of the anomaly detection. We utilize a variety of machine learning classifiers for anomaly detection, including *KitNET* (Mirsky et al. [2018]), an ensemble of autoencoders; *Multi-Layer Perceptron (MLP)* for deep learning; *Logistics Regression (LR)*, *Decision Tree (DT)*, and *Support Vector Machine (SVM)* for traditional approaches; and *Isolation Forest (IF)* for outlier detection. Our experiments cover 6 types of attacks—*Man-in-the-Middle (MITM)*, *Botnet*, *Brute Force*, *DDoS*, *Port Scan*, and *Infiltration*—utilizing data from Kitsune and the CICIDS-2017 dataset.

**Evaluation**. To assess the effectiveness of the mimicked packets in evading the 6 classifiers, we employed a singular, highly illustrative metric from the Traffic Manipulator: the Malicious traffic Evasion Rate (MER). This metric is calculated using the formula: $MER = 1 - \left(N^{\mathrm{adv}}/N^{\mathrm{mal}}\right)$, where $N^{\mathrm{adv}}$ and $N^{\mathrm{mal}}$ represent the number of detected adversarial and detected malicious packet sequences, respectively. This equation captures the percentage of adversarial packet sequences that goes undetected as compared to the original malicious packet sequences that was detected, effectively measuring the evasion capability of the adversarial packet sequences.

We employed the two-sample Kolmogorov-Smirnov (K-S) test, a powerful non-parametric method used to determine the probabilistic differences between two data samples. Specifically, we compare the empirical cumulative distribution functions (eCDFs) of both the original malicious packets and the adversarial packets generated by our model. The K-S test calculates the maximum distance (K-S statistic, *D*) between these two eCDFs. This statistic measures the greatest deviation between the distribution of our adversarially modified packets and the original malicious packets (Hore et al. [2023]; Gretton et al. [2012]). A *D* value exceeding the critical threshold at a chosen significance level suggests a rejection of the null hypothesis—that is, the two samples are not derive from the same distribution. We demonstrated adversarial packets that were found to be out-of-distribution (OOD) at 95% significance level.

Additionally, we used Wireshark, a widely recognized network protocol analyzer to ensure that during the modifications were made, the immutable fields crucial for the packet's attack functionality were preserved. This visual verification via Wireshark confirms that our adversarial packets maintain their structural integrity and functionality, successfully mimicking genuine network traffic while evading detection (Due to limited space, results and analysis are in Appendix B.1).

We also evaluated DiffuPac's efficacy in generating adversarial packets that maintain their malicious functionality in a controlled test environment using UTM hypervisor technology. This setup included two virtual machines: one running Kali Linux (attacker) and another running Ubuntu 24.04 (victim). The isolation ensured a realistic yet controlled network environment. We conducted the evaluation on two type of attacks: *Port Scan* and *Brute Force*. (Due to space constraints, results and analysis of the two attacks are in Appendix B.3.1 and B.3.2).

## 4.2 Results and Analysis

**Evasion Rate**. We evaluated the evasion rates of the 6 attacks using DiffuPac and two baseline models, and compared the results in Table 2 (Due to space constraints, the other 4 attacks are demonstrated in Appendix B.2). As shown in Table 2, we conclude that DiffuPac is capable of generating adversarial packets that achieves comparable or even higher in evasion rate compared with both Traffic Manipulator (GAN & PSO) and TANTRA (LSTM). Importantly, DiffuPac achieves this high performance without relying on NIDS components, surrogate classifiers or specific insights into the NIDS's feature extraction methods, highlighting its effectiveness in realistic settings where attackers lack access to such insider information. The evasion performance varied significantly across different classifiers, reflecting the inherent variability in their robustness and detection capabilities.

Flow-based NIDS exhibited greater resilience against our attacks compared to packet-based NIDS, likely due to DiffuPac's focus on packet-level modifications. Interestingly, the Traffic Manipulator model performed well with KitNet's AfterImage feature extractor, likely because it was trained directly on these features, enhancing its capability to modify them to evade detection. This specific alignment with NIDS features, while effective, does not typically reflect real-world attacker capabilities or constraints.

Table 3 : Percentage of successful adversarial samples found to be OOD.

| Attack Type | Percentage of OOD Samples (%) |
|---|---|
| Botnet | 48.72 |
| MITM | 29.34 |
| Port Scan | 55.23 |
| DDoS | 78.14 |
| Infiltration | 58.06 |
| Brute Force | 69.67 |

TANTRA also demonstrated high evasion rates, particularly with AfterImage, thanks to its use of LSTM models trained on benign traffic from the targeted network. This training allows TANTRA to reshape the timing between packets. This specific adjustment of interpacket delays directly impacts the time-based features extracted by AfterImage, making it more difficult for the classifiers to distinguish between normal and malicious packets.

Quantitatively, DiffuPac outperformed Traffic Manipulator by an average of 9.12 percentage points and TANTRA by an average of 4.26 percentage points across all attack types. The overall average improvement of DiffuPac over both baselines is 6.69 percentage points, demonstrating its superior capability to generate adversarial packets that effectively evade detection.

Based on the results of all the attacks, we can conclude that ML classifier are more robust than DNNs to adversarial attacks. Traditional ML models, like DT and IF, have simpler and more interpretable decision boundaries, making them less susceptible to subtle adversarial modifications (Sauka et al. [2022]). In contrast, DNNs with their complex and high-dimensional decision boundaries are more easily misled by the nuanced modifications introduced by DiffuPac.

**Statistical Difference (K-S test)**. The percentage of successful adversarial packets that were found to be OOD are demonstrated in Table 3. From the result, we can obeserve that Brute Force and DDoS attacks show the highest percentage of OOD samples. The grounded reasons for these kind of attacks are due to the inherent nature of their traffic patterns. The nature of Brute Force attacks involves numerous failed login attempts, leading to highly irregular sequence numbers (Javed and Paxson [2013]). Even after modifications by DiffuPac, these attempts will stand out due to their frequency and pattern, resulting in significant deviations in the eCDFs. The sheer volume and repetitive nature of DDoS traffic (Haseeb-ur rehman et al. [2023]) make it difficult to disguise effectively. Despite efforts of intelligent mutation process, the high volume and distinctive traffic patterns will cause substantial deviations in the eCDFs as well. On the other hand, the MITM show the lowest percentage of OOD samples. Adjusting TTL values is straightforward as they can be set to typical ranges seen in normal packets without significantly altering packet behavior, resulting in minimal deviation in the eCDFs.

## 5 Limitations

Our evaluations demonstrated DiffuPac's strong evasion capabilities, enabling the generation of adversarial packets that convincingly mimic legitimate network traffic, particularly in packet header fields. While these results highlight DiffuPac's ability to evade detection by NIDS, there remain areas for further study.

In particular, while we assessed the malicious functionality of generated adversarial packets in a controlled environment using UTM hypervisor technology, this evaluation was limited to *Port Scan* and *Brute Force* attacks. Although DiffuPac successfully retained the malicious intent of these attacks, further evaluations are required to determine its effectiveness across a broader range of attacks, especially those with more complex behaviors such as DDoS or Botnet attacks.

We also noted that DiffuPac's evasion performance is not uniform across all scenarios, notably DDoS and Brute Force. This variability is due to the distinct nature of each attack's traffic patterns, which may not be fully captured by the model, especially in highly repetitive or anomalous behaviors. To enhance the model's adaptability and generalizability, we plan to enrich the training dataset with a broader spectrum of real-world attack scenarios.

Moreover, the dual-use nature of adversarial generation models like DiffuPac presents significant ethical and legal challenges. While designed to improve security defenses, these technologies could be misused for malicious activities, such as facilitating cyber-attacks or disrupting services. To mitigate these risks, we have opted not to publicly release saved model checkpoints, aiming to prevent exploitation by malicious entities and ensure that our advancements in adversarial packet generation are used responsibly and within ethical bounds.

# 6    Conclusion and Future Directions

In this study, we introduced DiffuPac, an intelligent generative model that successfully generates adversarial packets capable of evading advanced NIDS while maintaining attack functionality. Unlike previous research, which often assumes attacker access to NIDS, DiffuPac operates under constraints of limited attacker knowledge, reflecting more realistic scenarios. Here, we present the in-depths insights obtained from this study: (a) DiffuPac uniquely combines normal and malicious packet sequences using contextual alignments, ensuring seamless integration into genuine traffic while employing these normal packet sequences to guide the denoising process. Through the performance evaluations with various NIDS, our model achieved an average improvement of approximately *6.69* percentage points in evasion rate compared to the two baselines across all attack types. (b) Evasion rates varied notably across attack types, with DDoS and Brute Force attacks showing higher probabilistic differences due to their complex, repetitive nature. This highlights potential areas for further refinement in DiffuPac's approach to handling voluminous attacks. (c) Simpler ML models like DT and IF displayed surprising resilience due to their less complex decision boundaries, limiting the effectiveness of DiffuPac's modifications. In contrast, DNNs, with their intricate decision boundaries, were more vulnerable, underscoring the complexities inherent in designing robust adversarial tactics. (d) DiffuPac's capability to balance sophisticated attacks with operational stealth makes it especially suitable for environments where attackers lack comprehensive NIDS configurations, enhancing its utility in realistic defense testing.

As future studies, we will compare DiffuPac against a broader generative adversarial packets model and extend testing across more diverse NIDS to better understand its relative strengths and limitations. Next, we will further expand the range of attack types included in the malicious functionality evaluation. Moreover, adversarial defenses are a key focus of our ongoing research, particularly in relation to DiffuPac's capabilities. One promising direction is the development of adversarial purification using diffusion models, a technique successfully applied in image processing. This approach treats adversarial perturbations as noise, utilizing the diffusion model's reverse process to restore network packets to their original state before analysis by NIDS. We believe this method has significant potential for enhancing network security and aim to further develop and test it as a novel defense strategy.

# Acknowledgements

This work has been partially supported by KAKENHI Grant Number JP23H03379.

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

# A    Model Architecture

## A.1    BERT

The core of DiffuPac is the BERT model, specifically adapted to serve as the denoising component in the diffusion process. The architecture consists of 12 bidirectional Transformer blocks and each block incorporates 12 attention heads within its self-attention layers (Vaswani et al. [2017]). These layers are crucial for capturing the intricate and implicit relationships between traffic bytes. Each input token has a dimension of 768, and the model can process sequences with up to 512 tokens. This setup allows the BERT model to effectively handle extensive sequences of traffic data, making it well-suited for the task of denoising during the diffusion process.

To implement of our pre-trained BERT model, we utilized the codebase from the UER (Universal Encoder Representations)-py project (Zhao et al. [2019]). UER-py is a versatile toolkit designed for pre-training and fine-tuning various NLP models, including BERT. In UER-py, the MLM is designed to predict masked words within a sentence. We adapted this approach to our masked unidirectional flow model, which focuses on predicting masked tokens within network traffic sequences. Instead of textual tokens, our model deals with traffic units from src-to-dst and dst-to-src sequences. We modified the MLM implementation to handle these traffic tokens, ensuring that each token is masked with a probability of 15%. Our model then predicts the original token from the masked sequence, capturing semantic patterns within bidirectional network flows.

The Next Sentence Prediction (NSP) task in UER-py involves prediction step to determine whether one sentence follows another. We transformed this concept into the same sequence-origin prediction task to determine the origin of directional network traffic. We adapted the binary classifier used in NSP to classify whether packets are originated from src-to-dst or dst-to-src. By pairing packets and predicting the directional flow, our model learns the contextual relationships and patterns in network traffic, similar to how NSP learns sentence relationships.

Table 4 lists the parameters are set to run our pre-trained BERT model. The remaining parameters were set to their default values.

We also conducted a comprehensive evaluation of different BERT model configurations to identify the most efficient option that still maintains high performance. We tested four BERT model variants: Tiny, Small, Medium, and Large, comparing their specifications, training time, loss, and accuracy. The results are summarized in Table 5. The Medium BERT model was chosen as the optimal configuration for DiffuPac because it offers a balanced trade-off between accuracy and loss metrics.

Table 4: Parameters used for pre-training the BERT model.

| Parameter | Value |
|---|---|
| Embedding Size | 768 |
| Feedforward Size | 3072 |
| Hidden Size | 768 |
| Hidden Activation Function | GELU |
| Number of Attention Heads | 12 |
| Number of Transformer Layers | 12 |
| Max Sequence Length | 512 |
| Dropout Rate | 0.1 |
| Batch Size | 64 |
| Total Steps | 500000 |
| Learning Rate | 1e-4 |

## A.2    Fine-tuning with Diffusion Model

For the fine-tuning phase of our model, we leveraged the DiffuSeq codebase (DiffuSeq's code). The DiffuSeq framework, originally designed for text data, required several significant adaptations to handle network traffic data and to meet the objectives of our diffusion model for adversarial packet generation. The primary adaptation to the DiffuSeq codebase involves the data preparation phase,

Table 5: Comparison of BERT Models.

| Parameters / Metrics | Tiny BERT | Small BERT | Medium BERT | Large BERT |
|---|---|---|---|---|
| **Model Specifications** | | | | |
| Embedding Size | 128 | 512 | 768 | 1024 |
| Feedforward Size | 512 | 2048 | 3072 | 4096 |
| Hidden Size | 128 | 512 | 768 | 1024 |
| Activation Function | GELU | GELU | GELU | GELU |
| Number of Heads | 2 | 8 | 12 | 16 |
| Number of Layers | 2 | 4 | 12 | 24 |
| **Training** | | | | |
| Training Time (hours) | 13.5 | 18 | 23 | 30.5 |
| **Performance** | | | | |
| MUM Loss | 1.028 | 0.822 | 0.667 | 0.610 |
| SSP Loss | 0.295 | 0.204 | 0.099 | 0.061 |
| MUM Accuracy | 0.844 | 0.887 | 0.905 | 0.914 |
| SSP Accuracy | 0.846 | 0.902 | 0.972 | 0.982 |

specifically the concatenation strategy used to blend normal and malicious packet sequences. This adaptation is crucial for the model to learn how to generate adversarial packets that can mimic normal packets and evade detection by NIDS.

In our concatenation strategy, we identified pairs of normal and malicious packet sequences that exhibit strong contextual alignments unlike DiffuSeq, which, randomly merge the text sequences. This means that the patterns and behaviors within these sequences are similar, ensuring that the concatenated sequences are blended seamlessly. We concatenated these contextually aligned normal and malicious packet sequences to form a new sequence, $\mathbf{S}^{\text{ben}\oplus\text{mal}}$. The strategy leverages the inherent patterns and behaviors within normal packet sequences, which is crucial for the mimicking process of malicious packet sequences.

We utilized the pre-trained BERT embeddings in our concatenation strategy. BERT is designed to capture deep semantic relationships within sequences. This design makes BERT making it highly effective for analyzing complex patterns in network traffic. By employing the pre-trained embeddings, DiffuPac can accurately transform discrete packet data into continuous feature spaces, capturing the intricate dependencies between packets. This transformation is critical for the subsequent similarity analysis and pairing process. The pre-trained embeddings offer a rich representation of the packet sequences, so that the model can discern subtle contextual similarities between normal and malicious packet sequences. The detailed of the concatenation strategy is demonstrated in Algorithm 1.

Here, we set the similarity threshold, $\epsilon$ to 0.9. This is a strategic decision aimed at balancing the need for high contextual alignment with the practicalities of ensuring a sufficient number of matched pairs. Setting $\epsilon$ to 0.9 indicates that we are looking for pairs of packet sequences that are highly similar, capturing nearly all the essential contextual information shared between normal and malicious packet sequences. This high threshold ensures that the concatenated sequences are blended seamlessly, maintaining the realistic traffic patterns and behaviors required to evade detection by NIDS. From a practical standpoint, a threshold of 0.9 is chosen because it strikes an optimal balance between precision and recall. A higher threshold (closer to 1.0) would result in fewer matched pairs, as only the most similar sequences would be selected. This could limit the model's ability to generate a diverse set of adversarial packets. Conversely, a lower threshold would increase the number of matched pairs but might include sequences that are not contextually well-aligned, reducing the effectiveness of the adversarial packets in mimicking normal traffic.

During the reverse process, we introduced a guidance mechanism using normal packet sequences. This mechanism treats the concatenated sequences of normal and noise-added malicious packets as a unified entities, allowing the model to denoise the malicious packets in the context of the normal packet sequences. We recalibrated the variational lower bound $L_{VLB}$ to emphasize the integration of normal sequence guidance. The objective focuses on accurately reconstructing the initial state, $\mathbf{z}_0$ from the noised states, so that the malicious components are seamlessly integrated into the normal packet sequences pattern.

Table 6 lists the parameters used to run our diffusion model. The other parameters are set according to the default values of DiffuSeq. All training, experimentation, and sampling processes are executed on a single NVIDIA AD102 (GeForce RTX 4090) GPU. The training process for BERT required approximately 23 hours, while fine-tuning with the diffusion model took roughly 10 hours.

Table 6: Parameters used for the diffusion model.

| Parameter | Value |
|---|---|
| Diffusion Steps | 2000 |
| Learning Rate | 1e-4 |
| Learning Steps | 50000 |
| Seed | 102 |
| Noise Schedule | sqrt |
| Batch Size | 64 |
| Microbatch | 64 |
| Sequence Length | 128 |
| Hidden Time Dimension | 128 |
| Hidden Dimension | 128 |
| Schedule Sampler | uniform |

---

**Algorithm 1** Finding contextually relevant packet sequences.

---

**Require:** Normal packet sequences $\mathbf{S}^{\mathrm{ben}}$, Malicious packet sequences $\mathbf{S}^{\mathrm{mal}}$, Model parameters $\theta$, Similarity threshold $\epsilon$
**Ensure:** Matched pairs of normal and malicious packet sequences $(\mathbf{S}_i^{\mathrm{ben}}, \mathbf{S}_j^{\mathrm{mal}})$
1: **Initialize:** Load pre-trained BERT model with parameters $\theta$
2: **Embed Normal Sequences:**
3: **for** each sequence $\mathbf{S}_i^{\mathrm{ben}}$ in $\mathbf{S}^{\mathrm{ben}}$ **do**
4:     Compute embedding $\mathbf{E}_i^{\mathrm{ben}} \leftarrow \mathrm{EMB}(\mathbf{S}_i^{\mathrm{ben}}; \theta)$
5: **end for**
6: **Embed Malicious Sequences:**
7: **for** each sequence $\mathbf{S}_j^{\mathrm{mal}}$ in $\mathbf{S}^{\mathrm{mal}}$ **do**
8:     Compute embedding $\mathbf{E}_j^{\mathrm{mal}} \leftarrow \mathrm{EMB}(\mathbf{S}_j^{\mathrm{mal}}; \theta)$
9: **end for**
10: **Find Contextually Aligned Pairs:**
11: **for** each embedding $\mathbf{E}_j^{\mathrm{mal}}$ in $\mathbf{E}^{\mathrm{mal}}$ **do**
12:     Initialize best match $b_j \leftarrow$ None
13:     Initialize highest similarity $\sigma_{\max} \leftarrow -1$
14:     **for** each embedding $\mathbf{E}_i^{\mathrm{ben}}$ in $\mathbf{E}^{\mathrm{ben}}$ **do**
15:         Compute similarity $\sigma_{ij} \leftarrow \mathrm{cosine\_similarity}(\mathbf{E}_j^{\mathrm{mal}}, \mathbf{E}_i^{\mathrm{ben}})$
16:         **if** $\sigma_{ij} > \sigma_{\max}$ **then**
17:             Update best match $b_j \leftarrow \mathbf{S}_i^{\mathrm{ben}}$
18:             Update highest similarity $\sigma_{\max} \leftarrow \sigma_{ij}$
19:         **end if**
20:     **end for**
21:     **if** $\sigma_{\max} > \epsilon$ **then**
22:         Add pair $(b_j, \mathbf{S}_j^{\mathrm{mal}})$ to matched pairs
23:     **end if**
24: **end for**
25: **Return Matched Pairs:** Return all matched pairs $(\mathbf{S}_i^{\mathrm{ben}}, \mathbf{S}_j^{\mathrm{mal}})$

---

# B Experimental Results

## B.1 Wireshark Results

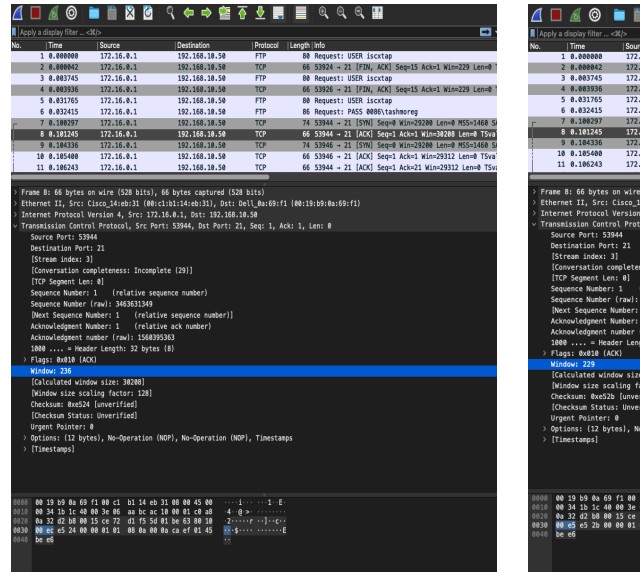

(a) Botnet attack before in Wireshark.    (b) Botnet attack after in Wireshark.

Figure 2: Comparison of Botnet attack before and after in Wireshark.

To demonstrate that our model intelligently alters specific fields to seamlessly blend malicious packets into normal traffic, we analyzed the packets using Wireshark. Our analysis covered 6 attack types: *MITM*, *Botnet*, *Brute Force*, *DDoS*, *Port Scan*, and *Infiltration*. The results showed that different specific fields were modified in the packets corresponding to each type of attack. This variation in field alteration confirms that our model adapts its modifications according to the nature of the attack, enhancing the stealthiness of the malicious packets. In Figure 2, we can see that there is a change in the window size of a Botnet packet. Botnet attack can be described as a network of infected devices (bots) controlled by an attacker to perform various malicious activities. Botnet traffic often shows abnormal patterns in window size due to automated control and data bursts. Here, we can conclude that DiffuPac made the intelligent decision by itself in modifying the window size to fall within typical user traffic patterns. This helps to camouflage the botnet activity.

MITM, on the other hand, demonstrated attacks whereby an attacker intercepts communication between two parties without their knowledge, often to eavesdrop or alter the data being sent. Abnormal TTL values (time-to-live field) can reveal this kind of interception and redirection. Here, once again, our model intelligently adjusts TTL value to match the expected values in aiding the masking of MITM presence. As shown in Figure 3, DiffuPac adjusts the TTL values to effectively conceal signs of interception. This ensures that the adversarial packets blend seamlessly with normal data exchanges, mimicking legitimate communication between parties.

DDoS attacks involve a large number of packets with similar characteristics, often reusing or predictably incrementing the IP Identification (ID) field. In Figure 4, it can be shown that DiffuPac addresses this by randomizing the IP IDs to prevent detection. When we analyzed the traffic using Wireshark, we observed varied and less predictable IP IDs in the modified packets, aligning with normal traffic patterns. This randomness helps avoid forming detectable patterns and evades detection systems that rely on identifying repetitive IP ID sequences. Thus, DiffuPac successfully masks the attack's presence while preserving the attack functionality.

Port Scan often involve specific flags, such as SYN, to probe for open ports. This behavior is distinct from normal traffic, which uses a variety of flags. In Figure 5, we found that DiffuPac diversifies the use of flags to include SYN, ACK, and FIN, similar to normal connections. By normalizing the traffic in this way, DiffuPac helps disguise scanning activity. For example, instead of sending many SYN

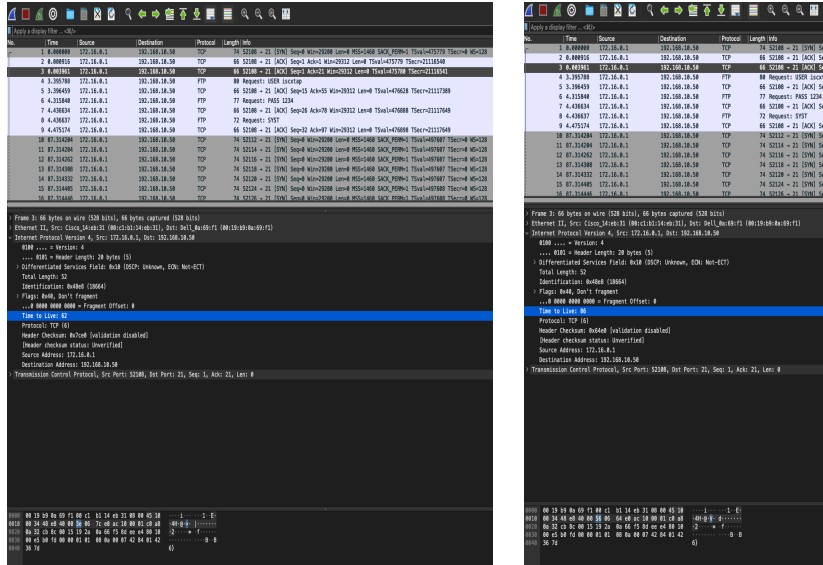

(a) MITM attack before in Wireshark.  (b) MITM attack after in Wireshark.

Figure 3: Comparison of MITM attack before and after in Wireshark.

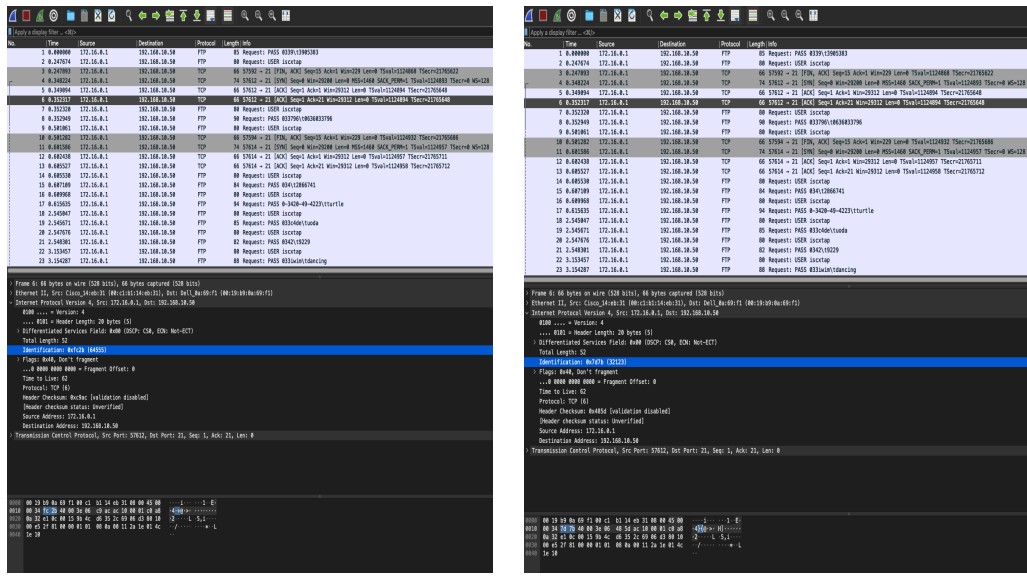

(a) DDoS attack before in Wireshark.  (b) DDoS attack after in Wireshark.

Figure 4: Comparison of DDoS attack before and after in Wireshark.

packets, it includes a mix of SYN, ACK, and FIN packets, which helps blend the scan into typical network behavior and evade detection.

Brute force attacks generate numerous login attempts, leading to irregular sequence numbers due to the repeated connections. By analyzing the traffic with Wireshark, we observed that DiffuPac intelligently normalizes sequence numbers to mimic regular user connections. Normal user login attempts typically show sequentially increasing sequence numbers. In Figure 6, it shows that DiffuPac adjusts these numbers to follow a similar pattern, thereby camouflaging the brute force activity. This adjustment helps blend the attack packets seamlessly into benign traffic, making it harder for NIDS to spot the abnormality.

Infiltration attacks involve moving laterally within a network, often generating unusual acknowledgment numbers as the attacker accesses various systems. In Figure 7, it demonstrated that DiffuPac

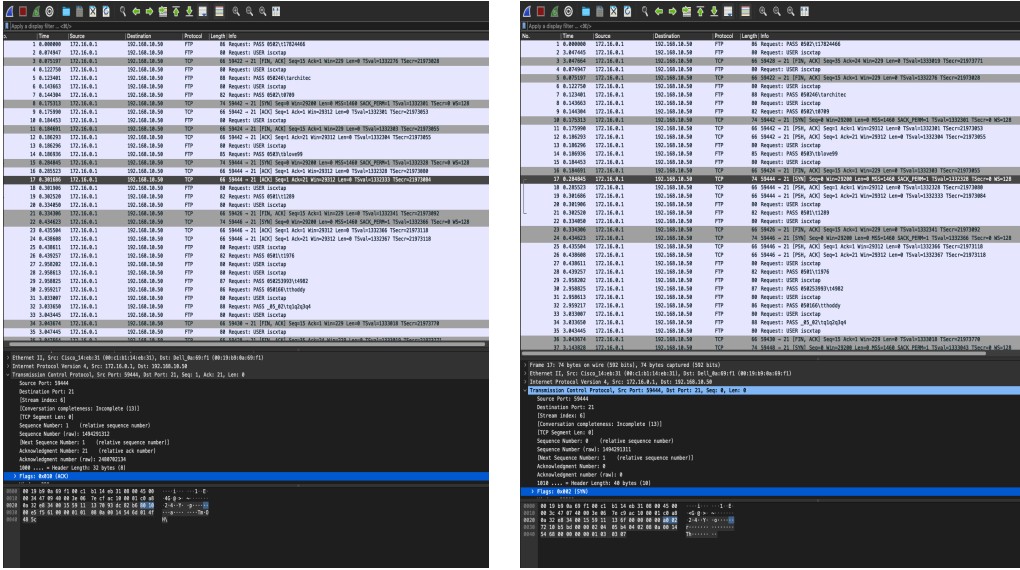

(a) Port Scan attack before in Wireshark.

(b) Port Scan attack after in Wireshark.

Figure 5: Comparison of Port Scan attack before and after in Wireshark.

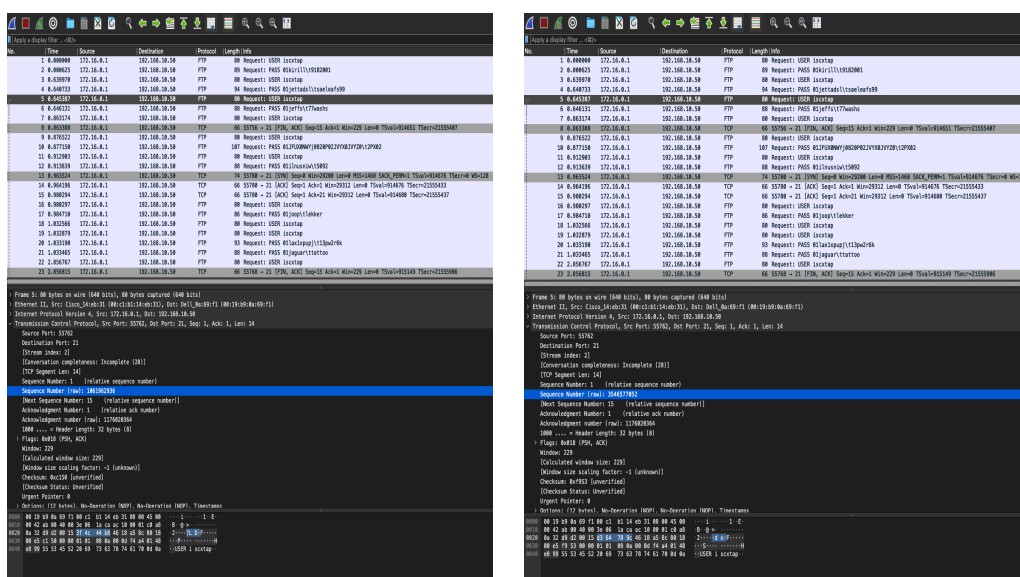

(a) BruteForce attack before in Wireshark.

(b) BruteForce attack after in Wireshark.

Figure 6: Comparison of bruteForce attack before and after in Wireshark.

intelligently modifies acknowledgment numbers to follow expected sequences. In normal communication, acknowledgment numbers increment predictably based on the received data. DiffuPac ensures that infiltration attempts have acknowledgment numbers that follow these normal patterns, effectively masking the lateral movement within the network and blending the attack traffic with normal traffic, making it difficult for detection systems to identify the anomaly.

DiffuPac demonstrates its ability to modify specific fields intelligently in the packet headers of different attack types, such as the window size for botnet traffic and TTL for MITM, to blend malicious packets into normal packets. This capability is consistently proven through detailed packet analysis with Wireshark. Indeed, the detailed packets analysis shows that critical fields defining packet identity and practicality (e.g., IP addresses, ports, payload) remain intact after modification, ensuring the functionality of the attack while evading detection by sophisticated NIDS. The comparisons

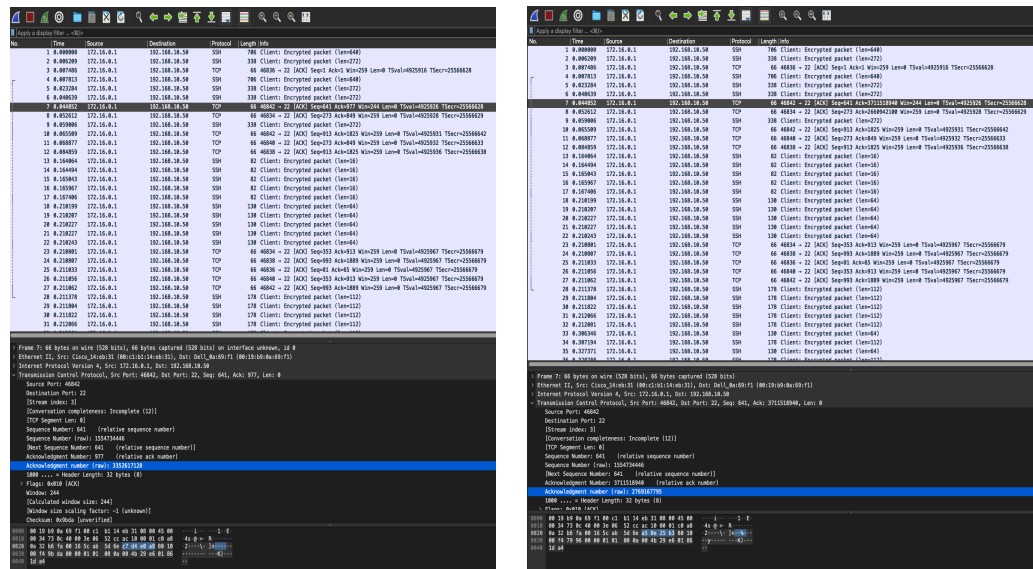

(a) Infiltration attack before in Wireshark.  (b) Infiltration attack after in Wireshark.

Figure 7: Comparison of infiltration attack before and after in Wireshark.

for each attack type before and after the modificatinos further highlight DiffuPac's effectiveness in maintaining attack practicality while enhancing stealth.

## B.2 Evasion Rate

We evaluated the evasion rates of the other 4 attacks—DDoS, Port Scan, Brute Force, and Infiltration—using DiffuPac and the baseline models, with the detailed results summarized in Table 7. According to the results, we can observed that the trends in the 4 attacks are consistent with the 2 attacks shown in Table 2 in the main paper.

Table 7: Comparative analysis of attack detection and evasion rates of the other 4 attacks.

**(c) DDoS**

| Feature Extractor | Classifier | Detection | | | Evasion (*MER*) | | |
|---|---|---|---|---|---|---|---|
| | | P | R | F1 | GAN & PSO | LSTM | **Ours** |
| CIC FLowMeter | KitNET | 0.94 | 0.90 | 0.92 | 33.19% | 41.69% | **52.06%** |
| | DT | 0.73 | 0.81 | 0.72 | 39.92% | 40.80% | **46.81%** |
| | IF | 0.99 | 0.91 | 0.93 | 26.74% | 39.49% | **51.45%** |
| | MLP | 0.72 | 0.71 | 0.74 | 52.03% | 48.17% | **69.87%** |
| | SVM | 0.73 | 0.77 | 0.78 | 42.87% | **58.47%** | 54.57% |
| | LR | 0.75 | 0.79 | 0.76 | 48.79% | 51.08% | **58.95%** |
| AfterImage | KitNET | 0.97 | 0.95 | 0.96 | 59.84% | **74.06%** | 62.31% |
| | DT | 0.78 | 0.92 | 0.85 | **72.88%** | 63.27% | 59.47% |
| | IF | 0.86 | 0.84 | 0.88 | 22.63% | 36.33% | **48.91%** |
| | MLP | 0.91 | 0.94 | 0.92 | 55.45% | 57.77% | **63.25%** |
| | SVM | 0.99 | 0.90 | 0.94 | 78.51% | **79.97%** | 68.93% |
| | LR | 0.94 | 0.97 | 0.91 | **71.68%** | 69.95% | 56.66% |

**(d) Port Scan**

| Feature Extractor | Classifier | Detection | | | Evasive (*MER*) | | |
|---|---|---|---|---|---|---|---|
| | | P | R | F1 | GAN & PSO | LSTM | **Ours** |
| CIC FLowMeter | KitNET | 0.96 | 0.91 | 0.94 | 33.46% | 40.04% | **55.03%** |
| | DT | 0.78 | 0.72 | 0.73 | 34.01% | **61.23%** | 59.25% |
| | IF | 0.99 | 0.90 | 0.94 | 22.73% | 29.49% | **45.26%** |
| | MLP | 0.76 | 0.71 | 0.74 | 59.72% | 57.56% | **67.92%** |
| | SVM | 0.73 | 0.77 | 0.78 | **68.91%** | 58.37% | 60.95% |
| | LR | 0.78 | 0.83 | 0.79 | 42.04% | 49.78% | **54.21%** |
| AfterImage | KitNET | 0.94 | 0.96 | 0.93 | **69.26%** | 64.97% | 55.87% |
| | DT | 0.77 | 0.91 | 0.82 | 69.78% | 62.01% | **70.47%** |
| | IF | 0.84 | 0.91 | 0.89 | 18.38% | 22.15% | **48.30%** |
| | MLP | 0.96 | 0.94 | 0.97 | 49.73% | 56.62% | **64.02%** |
| | SVM | 0.97 | 0.96 | 0.95 | 76.40% | **78.96%** | 75.81% |
| | LR | 0.95 | 0.92 | 0.90 | 67.29% | 70.07% | **73.94%** |

**(e) BruteForce**

| Feature Extractor | Classifier | Detection | | | Evasive (*MER*) | | |
|---|---|---|---|---|---|---|---|
| | | P | R | F1 | GAN & PSO | LSTM | **Ours** |
| CIC FLowMeter | KitNET | 0.97 | 0.94 | 0.97 | 36.94% | 43.13% | **50.32%** |
| | DT | 0.75 | 0.77 | 0.72 | 40.56% | **44.87%** | 42.79% |
| | IF | 0.91 | 0.90 | 0.94 | 27.01% | 39.14% | **41.64%** |
| | MLP | 0.88 | 0.85 | 0.83 | 52.32% | 64.58% | **67.18%** |
| | SVM | 0.80 | 0.81 | 0.72 | 47.31% | 51.54% | **56.74%** |
| | LR | 0.79 | 0.75 | 0.78 | 49.05% | 58.77% | **61.89%** |
| AfterImage | KitNET | 0.96 | 0.93 | 0.95 | 81.84% | **86.39%** | 69.31% |
| | DT | 0.82 | 0.79 | 0.78 | 41.07% | 43.56% | **55.07%** |
| | IF | 0.88 | 0.98 | 0.92 | 24.43% | 22.99% | **39.71%** |
| | MLP | 0.91 | 0.94 | 0.92 | 59.69% | 63.09% | **68.25%** |
| | SVM | 0.94 | 0.92 | 0.96 | **68.51%** | 61.79% | 58.93% |
| | LR | 0.97 | 0.93 | 0.94 | 61.68% | 57.25% | **64.06%** |

**(f) Infiltration**

| Feature Extractor | Classifier | Detection | | | Evasive (*MER*) | | |
|---|---|---|---|---|---|---|---|
| | | P | R | F1 | GAN & PSO | LSTM | **Ours** |
| CIC FLowMeter | KitNET | 0.91 | 0.90 | 0.96 | 41.71% | 40.68% | **45.04%** |
| | DT | 0.72 | 0.78 | 0.79 | 32.47% | 48.81% | **62.67%** |
| | IF | 0.99 | 0.91 | 0.94 | 28.10% | 42.19% | **45.88%** |
| | MLP | 0.77 | 0.71 | 0.74 | 50.09% | 53.68% | **63.14%** |
| | SVM | 0.74 | 0.76 | 0.79 | 62.29% | **64.51%** | 56.11% |
| | LR | 0.71 | 0.80 | 0.76 | 43.05% | 50.67% | **61.94%** |
| AfterImage | KitNET | 0.92 | 0.98 | 0.97 | **73.44%** | 70.58% | 51.26% |
| | DT | 0.77 | 0.91 | 0.82 | 69.23% | **72.32%** | 62.54% |
| | IF | 0.84 | 0.91 | 0.89 | 23.61% | 27.51% | **42.59%** |
| | MLP | 0.96 | 0.94 | 0.97 | 50.65% | 48.73% | **72.25%** |
| | SVM | 0.94 | 0.96 | 0.98 | **72.48%** | 68.95% | 64.73% |
| | LR | 0.93 | 0.92 | 0.95 | 66.04% | 65.93% | **68.37%** |

## B.3 Malicious Functionality Evaluation

**Experimental Setup**. We conducted a robust evaluation of both Port Scan and Brute Force attacks to assess their malicious functionality in controlled, isolated network environments. The primary goal of this evaluation was to determine whether adversarially modified packets retain their ability to perform malicious actions. Throughout the evaluation, we focused on analyzing the response of both the original and adversarial packets using Wireshark, carefully observing how the target system reacted to each type of attack.

### B.3.1 Port Scan

Before proceeding to evaluate the generated adversarial packets of DiffuPac, we first demonstrated a a successful Port Scan attack within our controlled environment. This initial step is a fundamental step to ensure the validity and reliability of subsequent tests with both original and adversarial packets. The network configurations depicted in Figure 8(a) and Figure 8(b) confirm that both virtual machines were correctly placed within the same subnet. As demonstrated in Figure 8(c), the Port Scan successfully identified open ports on the Ubuntu, highlighting port 22 as vulnerable.

We initiated a Port Scan attack from the Kali Linux machine targeting the Ubuntu server. During the execution of this attack, we captured all packets sent from the attacker to the victim, saving them as a pcap file. This capture was timed precisely to begin as the Nmap tool launched the Port Scan, ensuring that only the relevant src-to-dst packets were recorded. This focus aligns with the operational framework of DiffuPac, which concentrates on modifying packets sent from the source.

Using the captured src-to-dst packets, we employed DiffuPac to generate adversarial Port Scan packets, which were then saved in a separate pcap file. These adversarial packets were subsequently replayed using the Tcpreplay tool, directing them towards the Ubuntu server. Concurrently, we used Wireshark on the Ubuntu machine to capture the incoming responses, allowing us to analyze the effectiveness of the adversarial packets in real-time.

To ensure a balanced evaluation, we also replayed the original Port Scan pcap file using Tcpreplay, capturing the responses in Wireshark on the Ubuntu side. This dual replay enabled a direct comparison between the packets' responses generated by the original and adversarial Port Scan packets.

As depicted in Figure 9, the response to the adversarial Port Scan packets closely mirrors that of the original Port Scan. A detailed inspection of the TCP flags reveals that packets with SYN and ACK flags targeted port 22, corroborating the initial vulnerability identified in Figure 8(c). This consistent response across both original and adversarial packets provides compelling evidence that DiffuPac successfully generates adversarial packets that retain their intended malicious functionality.

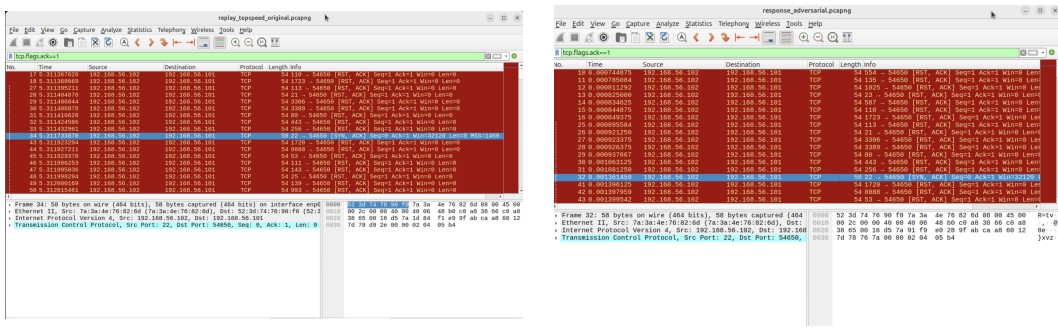

a) Kali Linux network configurations  b) Ubuntu network configurations

c) Successfully executed Port Scan using Nmap tool

Figure 8: Port scan attack demonstration on isolated environment

a) Original Port Scan Response  b) Adversarial Port Scan Response

Figure 9: Comparison of Victim Response demonstrated in wireshark

### B.3.2 Brute Force

For the Brute Force attack evaluation, the experimental setup followed a similar approach to that of the Port Scan, focusing on capturing and analyzing both original and adversarial packets. However, in this case, we leveraged Metasploit's ssh_login module to perform the attack. After demonstrating a successful Brute Force attack using Metasploit, we captured the relevant packets and utilized DiffuPac to generate adversarial versions of those packets. The evaluation focused on comparing the response of original and adversarial packets, observing how successful logins were achieved in both cases.

As shown in Figure 10, we utilized the Metasploit Framework to perform a Brute Force attack targeting the SSH service on the Ubuntu server. After setting the RHOSTS to the Ubuntu machine, we provided both a user file and a password file containing potential login credentials. The ssh_login module systematically attempted each combination of usernames and passwords, ultimately leading to a successful login.

The success of the Brute Force attack is demonstrated in Figure 10, where Metasploit successfully opened an SSH session on the target machine. This provided the necessary confirmation that the attack could be executed under normal conditions, laying the foundation for the subsequent evaluation of adversarial packets.

Following the successful attack, similar to the Port Scan setup, we captured the src-to-dst packets exchanged during the Brute Force attack. These packets were saved as a pcap file, which was subsequently used as input for DiffuPac to generate adversarial Brute Force packets.

Using Tcpreplay, both the original and adversarial Brute Force packets were replayed towards the Ubuntu server. As shown in Figure 11, the response to the adversarial Brute Force packets closely mirrored that of the original Brute Force packets. The packet exchange between the attacker and the target server increased significantly as the correct login credentials were identified, culminating in a successful SSH session for both the original and adversarial attacks.

To further analyze and validate the outcomes of both the original and adversarial Brute Force attacks, we utilized pam_unix(sshd) on the Ubuntu server to log and compare successful login attempts. As depicted in Figure 12, the log entries confirmed that the SSH service recorded successful logins for both the original Brute Force attack and the adversarial Brute Force attack.

Figure 10: Brute Force Attack Successfully conducted using metasploit

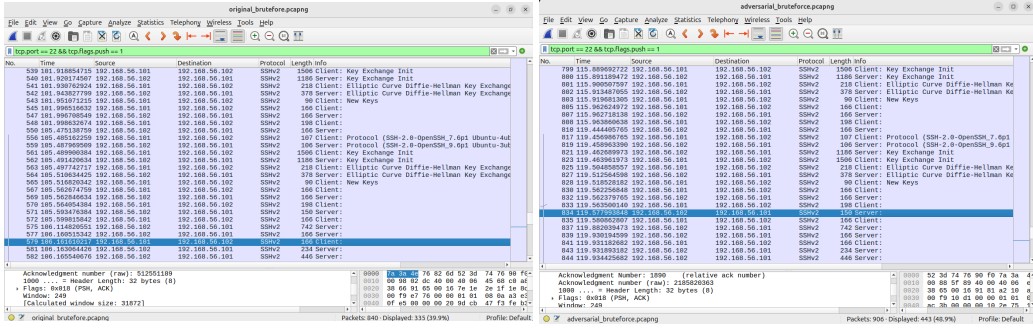

a) Original Brute Force           b) Adversarial Brute Force

Figure 11: Response Comparison of Original Brute Force and Adversarial Brute Force in Wireshark

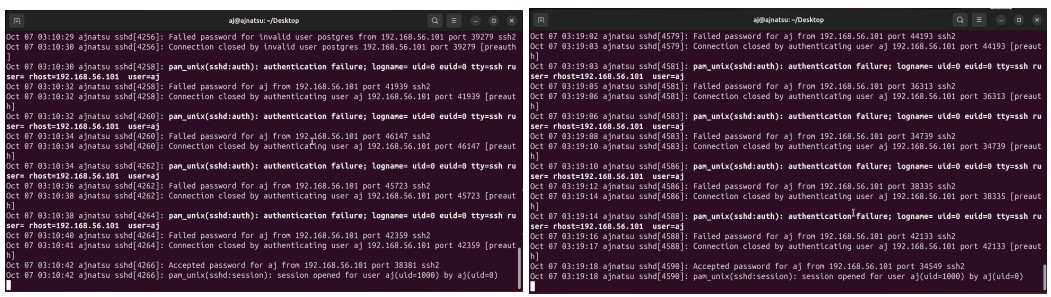

a) Original Brute Force           b) Adversarial Brute Force

Figure 12: Response Comparison of Original Brute Force and Adversarial Brute Force in ssh login

## C  Ablution Study

This ablation study assesses the impact of substituting the pre-trained BERT model with a standard transformer architecture in the denoising process of DiffuPac. The focus is on assessing each model's ability to accurately reconstruct and generate adversarial packets that can effectively evade detection.

**Non-BERT-based.** This variant of DiffuPac employs a standard transformer architecture for the denoising process, as outlined in the DiffuSeq methodology (Gong et al. [2023]). Notably, this model is trained from scratch without leveraging any pre-trained weights, thus relying solely on the information provided during the training phase.

**BERT-based.** The existing DiffuPac configuration, which incorporates a pre-trained BERT model for the denoising process, was utilized. This setup takes advantage of BERT's pre-trained weights, enabling enhanced semantic understanding and contextual relevance during packet reconstruction.

Both models were trained under identical conditions and evaluated based on their success in generating adversarial packets that evade detection. The results, as depicted in Table 8 , highlight the comparative effectiveness of each model.

As shown in Table 8, the BERT-based DiffuPac significantly outperforms the non-BERT variant, demonstrating superior ability to reconstruct packets that blend seamlessly into normal traffic. The BERT-based model excels in pairing malicious packets with normal packets that exhibit high contextual relevance. This strategic pairing allows the generated adversarial packets to blend seamlessly into normal traffic, thereby increasing their likelihood of evading detection. In contrast, the non-BERT-based model, which lacks this pre-trained knowledge, resorts to random matching of malicious and normal packets, leading to a noticeable reduction in the effectiveness of the generated adversarial traffic.

An intriguing aspect of DiffuPac is its dual utilization of the pre-trained BERT model: first, as the denoising engine during the packet reconstruction process, and second, as a critical tool for identifying contextually relevant packet pairs. To the best of our knowledge, this dual application of a

Table 8: Comparative analysis of attack detection and evasion rates.

(a) Botnet

| Feature Extractor | Classifier | Detection | | | Evasive (*MER*) | |
|---|---|---|---|---|---|---|
| | | P | R | F1 | Non-BERT-based | BERT-Based |
| CIC FLowMeter | KitNET | 0.84 | 0.94 | 0.92 | 38.93% | **46.48%** |
| | DT | 0.79 | 0.91 | 0.82 | 50.24% | **60.98%** |
| | IF | 0.99 | 0.90 | 0.95 | 39.18% | **49.71%** |
| | MLP | 0.92 | 0.84 | 0.86 | 53.85% | **63.05%** |
| | SVM | 0.99 | 0.92 | 0.95 | 52.31% | **64.49%** |
| | LR | 0.84 | 0.91 | 0.89 | 35.02% | **42.08%** |
| AfterImage | KitNET | 0.96 | 0.90 | 0.94 | 69.15% | **74.46%** |
| | DT | 0.80 | 0.90 | 0.82 | 65.77% | **72.13%** |
| | IF | 0.99 | 0.92 | 0.94 | 46.83% | **52.79%** |
| | MLP | 0.96 | 0.96 | 0.95 | 52.04% | **64.92%** |
| | SVM | 0.99 | 0.90 | 0.94 | 60.19% | **69.19%** |
| | LR | 0.97 | 0.90 | 0.93 | 53.98% | **58.98%** |

(b) MITM

| Feature Extractor | Classifier | Detection | | | Evasive (*MER*) | |
|---|---|---|---|---|---|---|
| | | P | R | F1 | Non-BERT-based | BERT-Based |
| CIC FLowMeter | KitNET | 0.92 | 0.94 | 0.91 | 45.24% | **55.87%** |
| | DT | 0.74 | 0.79 | 0.76 | 47.83% | **64.12%** |
| | IF | 0.99 | 0.92 | 0.94 | 36.91% | **52.99%** |
| | MLP | 0.77 | 0.72 | 0.74 | 54.98% | **73.21%** |
| | SVM | 0.74 | 0.79 | 0.78 | 56.10% | **60.46%** |
| | LR | 0.73 | 0.78 | 0.72 | 35.27% | **47.48%** |
| AfterImage | KitNET | 0.94 | 0.96 | 0.93 | 52.55% | **58.48%** |
| | DT | 0.75 | 0.89 | 0.84 | 61.47% | **70.04%** |
| | IF | 0.81 | 0.83 | 0.86 | 31.33% | **45.71%** |
| | MLP | 0.92 | 0.90 | 0.93 | 64.17% | **71.45%** |
| | SVM | 0.99 | 0.90 | 0.94 | 56.44% | **66.53%** |
| | LR | 0.91 | 0.94 | 0.90 | 44.83% | **52.38 %** |

pre-trained model within a diffusion framework is unprecedented, further highlighting the innovation and effectiveness of our approach.

