# OpenReview forum: "DiffuPac: Contextual Mimicry in Adversarial Packets Generation via Diffusion Model"
_NeurIPS.cc/2024/Conference — NeurIPS 2024 poster_

### Official Review · Reviewer_rUa8 · 2024-07-03

**Soundness:** 3
**Presentation:** 3
**Contribution:** 3
**Rating:** 5
**Confidence:** 4

**Summary:**

This paper proposes an adversarial packet generation method, named DiffuPac, to evade detection. DiffuPac integrates BERT and a diffusion model to make the generated packets indistinguishable. To better fit the task, a concatenation strategy and a classifier-free approach are proposed. The experimental results are promising.

**Strengths:**

1.	This paper proposes a novel adversarial packet generation scheme to test the evasion detection ability of NIDS.
2.	The paper is well-written and easy to follow.
3.	The performance is good.

**Weaknesses:**

1.	The primary concern is the efficiency of DiffuPac. As a diffusion model is adopted to generate adversarial packet, it may be time-consuming. Can you conduct some experiments to validate the efficiency of your method?
2.	Can you also consider some defenses against such adversarial packets? For example, adversarial training (AT), feature selection (FS), and adversarial feature reduction (AFR) as shown in work [1]


[1] Han, Dongqi, et al. "Evaluating and improving adversarial robustness of machine learning-based network intrusion detectors." IEEE Journal on Selected Areas in Communications 39.8 (2021): 2632-2647.

**Questions:**

1.	I understand the effectiveness of DiffuPac in an experimental environment but in the real world, usually, an ML model is adopted to detect evasion for fast response requirements but not a DNN. As DNN is less robust to adversarial examples, will this affect the practicability of DiffuPac?

**Limitations:**

Mentioned in Sec.5.

---

> ### Author Rebuttal · Authors · 2024-08-05
>
> We greatly appreciate the reviewers’ constructive review and insightful suggestions regarding our paper.
> Our responses are given in a point-by-point manner for each comment.
>
> **W1.**
>
> We conducted a comprehensive evaluation of different BERT model configurations to identify the most efficient option that still maintains high performance.
> As for the diffusion model,  we have implemented several optimizations to improve the training and inference efficiency of the model.
> We tested four BERT model variants: Tiny, Small, Medium, and Large, comparing their specifications, training time, loss, and accuracy.
> The results are summarized in the Table 1 below:
>
> | **Parameters**          | **Metrics**            | **Tiny BERT** | **Small BERT** | **Medium BERT** | **Large BERT** |
> |-------------------------|------------------------|---------------|----------------|-----------------|----------------|
> | **Model Specifications**| **Embedding Size**     | 128           | 512            | 768             | 1024           |
> |                         | **Feedforward Size**   | 512           | 2048           | 3072            | 4096           |
> |                         | **Hidden Size**        | 128           | 512            | 768             | 1024           |
> |                         | **Activation Function**| GELU          | GELU           | GELU            | GELU           |
> |                         | **Number of Heads**    | 2             | 8              | 12              | 16             |
> |                         | **Number of Layers**   | 2             | 4              | 12              | 24             |
> | **Training**            | **Training Time (hours)**| 13.5         | 18             | 23              | 30.5           |
> | **Performance**         | **MUM Loss**           | 1.028         | 0.822          | 0.667           | 0.610          |
> |                         | **SSP Loss**           | 0.295         | 0.204          | 0.099           | 0.061          |
> |                         | **MUM Accuracy**       | 0.844         | 0.887          | 0.905           | 0.914          |
> |                         | **SSP Accuracy**       | 0.846         | 0.902          | 0.972           | 0.982          |
>
> **Table 1**: Comparison of Tiny, Small, Medium, and Large BERT models in terms of their specifications, training time, loss, and accuracy.
>
> The Medium BERT model was chosen as the optimal configuration for DiffuPac because it offers a balanced trade-off between accuracy and loss metrics.
>
> As for the diffusion model, we utilized FP16 (half-precision floating-point) for GPU computations. This approach significantly reduced the training time from 10 hours to 4 hours, nearly doubling the speed without sacrificing model performance. FP16 allows us to leverage the GPU's computational power more effectively, enabling faster iterations and reduced memory usage.
> During the sampling phase, we initially encountered issues with memory over-utilization, which was a significant bottleneck given our hardware limitations.
> To resolve this, we integrated the FAISS library, which is optimized for efficient similarity search and clustering. FAISS allowed us to handle the nearest neighbor search more efficiently, preventing memory overflow and ensuring smoother sampling operations. This optimization not only improved the stability of our sampling phase but also reduced the overall memory footprint of the model.
>
> All of the results including the fully-detailed explanation of them will be added in the revised paper.
>
> **W2.**
>
> Adversarial Defences is a critical area of research that we are actively exploring, particularly in the context of DiffuPac's unique capabilities.
>
> In addition to the more established defense strategies mentioned, we are currently developing a novel approach to defending against adversarial packets: adversarial purification using diffusion models.
> This technique has shown great promise in the field of image processing, and we believe it holds significant potential for network security as well.
>
> Our approach leverages the inherent strengths of diffusion models, which are ideally suited for the task of adversarial purification.
> By treating adversarial perturbations as noise, we utilize the diffusion model's reverse process to "purify" incoming network packets, effectively stripping away malicious modifications and restoring the packet to its original, clean state before it is analyzed by a NIDS.
>
> This research is still in progress, and we are excited about its potential. We aim to fully develop and test this method, positioning it as a powerful alternative to traditional adversarial defenses.
> As far as we are aware, this research work will be the first to apply adversarial purification to network packets, which we believe could set a new standard in the field.
>
> **Q1.**
>
> In future work, we will propose a hybrid approach that combines the strengths of DiffuPac with more robust and faster-responding machine learning (ML) models.
> In practice, DiffuPac could be used to generate adversarial examples during the training phase, which can then be used to harden simpler, more interpretable models like decision trees, random forests, or support vector machines through adversarial training.
> This approach allows for the creation of a robust detection system that can quickly respond to threats while benefiting from the advanced adversarial generation capabilities of DiffuPac.
>
> Additionally, while DiffuPac focuses on generating adversarial examples, the integration of adversarial purification, as discussed in our ongoing research, offers another layer of defense.
> This process can help mitigate the effects of adversarial examples on DNNs, thereby enhancing their robustness in real-world applications.

---

> > ### Comment · Reviewer_rUa8 · 2024-08-12
> >
> > Thanks for the author's response. However, only discussion is provided but lacking some experiment results to address my concern. Thus, I tend to keep my score.

---

> > > ### Author Response · Authors · 2024-08-13
> > >
> > > We deeply appreciate the reviewer's feedback and apologize for not fully addressing the concerns during the rebuttal phase.
> > >
> > > In the revised paper, we will thoroughly address all feedback by increasing the necessary experimental evidence. We are confident that the additional data and analysis will alleviate the reviewer's concerns.

---

### Official Review · Reviewer_BP27 · 2024-07-11

**Soundness:** 3
**Presentation:** 3
**Contribution:** 3
**Rating:** 6
**Confidence:** 2

**Summary:**

This paper proposes a novel strategy named DiffuPac to generate adversarial packets to bypass NIDS detection. DiffuPac encompasses two critical components including the BERT, which captures the semantic meaning of packets and facilitates the embedding and contextual paring. The other one is DiffSeq which makes the malicious part seamlessly integrated into the normal packet pattern thereby improving the evasion against detection. Experiments showcase that DiffuPac outperforms the baseline and is capable of breaking practical detection through simulation.

**Strengths:**

1. This paper is well-written, with a clear method description.

2. The proposed DiffPac is effective against bypassing detection, the proposed strategy seems reasonable with good explanation.

**Weaknesses:**

1. As stated in the limitation, whether the functionality of these adversarial packets is preserved is not verified. From my understanding, this point is quite important given the practical aspect of DiffPac. If the adversarial package can only bypass detection without functioning as intended, it would undermine the practical utility of DiffPac.

2. It would be beneficial to include additional analysis, such as evaluating the performance without applying contextual pairing and assessing the robustness against potential packet-level defenses. These analyses would provide further insights into the effectiveness and resilience of DiffPac.

**Questions:**

Please refer to weakness.

**Limitations:**

The authors have listed their limitations and have addressed most of them. One concern about functionality preservation is mentioned in the weakness part.

---

> ### Author Rebuttal · Authors · 2024-08-05
>
> We greatly appreciate the reviewers’ constructive review and insightful suggestions regarding our paper.
> Our responses are given in a point-by-point manner for each comment.
>
> **W1.**
>
> We conducted a detailed malicious functionality evaluation within a controlled, isolated network environment.
> We kindly ask the reviewer to refer to the "Malicious Functionality Evaluation" section in the main author rebuttal for a more detailed explanation.
>
> **W2.**
>
> We have conducted an ablation study that directly addresses the impact of contextual pairing, comparing DiffuPac’s performance with and without the use of the pre-trained BERT model for contextual pairing.
> This study has been included in our main author rebuttal, and we encourage the reviewer to refer to "Ablation Study" section in the main author rebuttal for detailed insights into how contextual pairing enhances the model’s effectiveness.
>
> We acknowledge that while Kitsune NIDS provides a solid foundation for evaluating packet-level defenses, it represents only one approach among many.
> To further strengthen DiffuPac’s robustness and address the reviewer's concerns, our future work will explore additional NIDS that utilize advanced packet-based feature extractors.
> By broadening our evaluation framework to include a variety of packet-level defenses, we aim to gain a deeper understanding of DiffuPac’s effectiveness across different NIDS architectures and feature extraction methodologies.

---

> > ### Comment · Reviewer_BP27 · 2024-08-09
> >
> > Thanks for the response. The 'Malicious Functionality Evaluation' refers to Lines 382 - 384. While I acknowledge the discussion in the Limitation section, preserving the intended malicious effect is essential to ensure the practicality of your method.

---

### Official Review · Reviewer_vp4b · 2024-07-12

**Soundness:** 3
**Presentation:** 3
**Contribution:** 3
**Rating:** 5
**Confidence:** 4

**Summary:**

The paper presents DiffuPac, a novel solution that integrates Bidirectional Encoder Representations from Transformers (BERT) and diffusion models to generate adversarial packets that can evade detection by sophisticated Network Intrusion Detection Systems (NIDS).
DiffuPac leverages the extensive contextual understanding provided by BERT, which has been trained on diverse datasets representing a wide range of network behaviors, along with the generative capabilities of diffusion models. This fusion results in a sophisticated adversarial tactic where the elements of the attack are seamlessly integrated into the network traffic, making them indistinguishable from legitimate data.
Extensive experimental evaluations on real-world datasets demonstrate that DiffuPac significantly outperforms existing methods in terms of evasion effectiveness, establishing new benchmarks for the generation of adversarial packets.

**Strengths:**

* Integrates BERT and diffusion models for adversarial packet generation

* Addresses limitations of traditional methods that rely on unrealistic attacker knowledge

* Introduces unique concatenation and targeted noising techniques for seamless blending of adversarial packets

* I really appreciate that authors consider many realistic attacker constraints (e.g., packet integrity, replayability, only control on src-to-dst flows), showing that the authors have solid domain knowledge in network security

**Weaknesses:**

* There are many unclear points about the high-level idea, methodology, and experiments; see them below. If provided with compelling evidence, I may increase my score.

**Questions:**

1. For the high-level idea, I admit that there have been attempts to transform packet bytes as tokens and use language models for traffic analysis (e.g., [a,b,c]). However, most of them (also including DiffuPac) do not discuss if and how this type of method can handle encrypted traffic. As you may know, modern encryption algorithms like AES can significantly reduce entropy and there will be no one-to-one relations between plaintext and ciphertext; so from my perspective, learning from the encrypted payload is meaningless. Some methods may only use the header fields for tokenization. As there lack of such details, I'm not sure how DiffuPac deals with this concern.
2. On page 5, line 199: "... this task employs a dedicated binary classifier to determine the directional origin of network traffic, specifically whether packets in a sequence originate from src-to-dst or dst-to-src." Though the idea seems reasonable, I'm curious will there be any cases where packets from both sides are very similar so that the task can be too difficult to converge (e.g., in P2P apps where there are no explicit servers or clients)?
3. On page 5, line 217: "the initial packets in a flow contain the most significant information, we limit our analysis to the first three packets of each heavy flow". Will the first three packets be too few? For TCP connections, the first three packets are typically just for handshakes so the model probably won't learn any useful information. Can the authors explain this, or give some experiments on this parameter?
4. Could the authors list the mutable and immutable fields in the appendix?
5. For experiment setup, one of my concerns is about the data preparation. Will the pre-trained model use malicious traffic for training? For one thing, the pattern of malicious traffic can be very abnormal so will that affect the BERT's understanding of normal traffic? For another, in practice, malicious traffic is much less than in the dataset so this setting can be somewhat impractical for a pre-trained model.
6. Another major weakness of the experiment is that there is only one baseline. However, there are other works on generating (malicious or generic) traffic data, including those generating on feature spaces ([d,e]) or generating on raw bytes ([a,b]). The authors may compare them to improve the experiment, or discuss them in related work and present their fatal limitations that hinder the direct comparison.
7. As there are two main components in DiffuPac (BERT and diffusion model), an ablation study on the pre-trained model is needed to show the necessity of using the pre-trained representations instead of some simpler representations.
8. In the limitations, I suggest that the authors discuss the implications of your method facing signature-based NIDS (e.g., Snort). You may conclude with a consideration of the current situation that encrypted traffic is the majority, but that will go back to question 1 again.
9. I like the authors providing the screenshots of Wireshark as parts of the results. One weakness is that the authors use their own domain knowledge to reveal the important packet fields that should be altered to avoid NIDS. However, the ML/DL-based NIDS may use some other features as the decision reasons which can be different from yours. I think the authors may use some global explanation models (e.g., [f,g]) to verify if the adversarial fields are really the features that NIDSes care.

[a] NetGPT: Generative Pretrained Transformer for Network Traffic, arxiv 2023

[b] TrafficGPT: Breaking the Token Barrier for Efficient Long Traffic Analysis and Generation, arxiv 2024

[c] Yet Another Traffic Classifier: A Masked Autoencoder Based Traffc Transformer with Multi-Level Flow Representation, AAAI 2023

[d] Flow-based network traffic generation using Generative Adversarial Networks, SIGCOMM 2022

[e] Knowledge Enhanced GAN for IoT Traffic Generation, WWW 2023

[f] AI/ML for Network Security: The Emperor has no Clothes, CCS 2022

[g] Interpreting Unsupervised Anomaly Detection in Security via Rule Extraction, NeurIPS 2023

**Limitations:**

Yes.

---

> ### Author Rebuttal · Authors · 2024-08-05
>
> We greatly appreciate the reviewers’ constructive review and insightful suggestions regarding our paper.
> Our responses are given in a point-by-point manner for each comment.
>
> **Q1.**
>
> We appreciate the reviewer's insightful observation regarding the challenges of handling encrypted traffic.
> In DiffuPac, we treat the payload as immutable, especially when encrypted.
> This decision is based on the complexities of modern encryption algorithms like AES, which significantly reduce entropy and break the one-to-one relationship between plaintext and ciphertext.
> Modifying encrypted payloads without understanding their structure risks rendering the packets non-functional or easily detectable.
> Our current focus is on modifying packet headers, which offer a safer and more navigable path for creating adversarial effects.
> Header fields provide ample opportunity for evasion tactics while preserving the packet's integrity and functionality.
> Future work may explore techniques for selective and safe payload modification, possibly leveraging advanced machine learning models to identify modifiable regions within encrypted payloads.
>
> **Q2.**
>
> While the task of distinguishing similar packets from both sides can indeed be challenging, our classifier has shown a strong ability to converge, even in these complex scenarios.
> This is largely due to the deep learning capabilities of the transformer architecture, which uses self-attention mechanisms to capture subtle contextual cues and dependencies within packet sequences.
> We trained the classifier using datasets like CICIDS2017 and Kitsune, which include scenarios with P2P traffic.
> These datasets provide the necessary complexity to train the model to differentiate between src-to-dst and dst-to-src flows, even in environments where the distinction is not immediately apparent.
> We plan to incorporate more diverse traffic scenarios in future work, including different types of encrypted and IoT traffic.
>
> **Q3.**
>
> We apologize for any confusion caused by the initial explanation and appreciate the reviewer's insightful observation.
> We will refine the explanation in the revised version to prevent any misunderstanding.
>
> To clarify, in our preprocessing stage, each session is split into unidirectional flows—specifically src-to-dst and dst-to-src sequences.
> This effectively means our analysis considers up to six packets per session: three from the src-to-dst flow and three from the dst-to-src flow.
>
> By analyzing the first three packets from each unidirectional flow, we capture the initial exchanges in both directions.
> This includes not just the TCP handshake but also the initial data packets, which often contain valuable information.
> Considering packets from both directions ensures a more complete view of the communication, enhancing the model's ability to learn meaningful patterns even from early-stage data.
>
> **Q4.**
>
> We cannot update the supplementary materials during the rebuttal period, in the revised paper we will provide fully-detailed explanation on mutable and immutable fields.
>
> **Q5.**
>
> The pre-training phase utilizes an unlabeled dataset comprising a mix of normal and malicious traffic.
> This allows BERT to learn general network traffic patterns and dependencies without bias toward specific traffic types.
> While malicious traffic can exhibit abnormal patterns, BERT's generalization capabilities ensure that these do not overshadow its understanding of normal traffic.
> The focus during pre-training is on capturing the overall structure of network flows rather than distinguishing between normal and malicious traffic.
>
> **Q6.**
>
> We acknowledge that including only one baseline may not provide a comprehensive comparison, especially in light of the existing body of work on generating packets.
> In future work, we will conduct a thorough comparison with a broader range of methods within the Packet-Level Attacks category [1] to better position DiffuPac within the field of adversarial packets generation.
>
> **Q7.**
>
> We kindly ask the reviewer to refer to the "Ablation Study" section in the main author rebuttal for a more detailed explanation.
>
> **Q8.**
>
> We appreciate the reviewer's suggestion to discuss the implications of DiffuPac when facing signature-based NIDS.
> The discussion regarding the relevance of signature-based NIDS on DiffuPac will be included in the revised paper.
> This will involve evaluating how well DiffuPac's techniques, such as header modification and traffic pattern manipulation, can evade detection by these systems.
> Future work will involve testing DiffuPac against a range of signature-based NIDS configurations and exploring enhanced header manipulation and potential payload modification techniques that respect encryption constraints.
>
>
> **Q9.**
>
> DiffuPac does not depend on predefined domain knowledge to reveal important packet fields.
> Instead, it intelligently alters specific fields within packets to seamlessly blend malicious packets into normal traffic, enhancing evasion capabilities.
> This auto-altering capability is a key innovation of DiffuPac. In our revised paper, we will provide more refined explanations to prevent any misunderstandings regarding DiffuPac's intelligent field alteration capabilities.
>
> We acknowledge that there may be differences between the packet fields modified by DiffuPac and the features ML/DL-based NIDS consider critical.
> This potential disparity arises because DiffuPac independently identifies and modifies fields based on the inherent characteristics of network traffic and attack types.
>
> In future work, we will use global explanation models, such as SHAP or LIME, in a controlled setting to gain insights into the influential features in NIDS detection processes.
> Based on these findings, we may enhance DiffuPac's targeting of significant features while maintaining its independent operational framework.
>
> [1] K. He et.al, Adversarial Machine Learning for Network Intrusion Detection Systems: A Comprehensive Survey

---

> > ### Comment · Reviewer_vp4b · 2024-08-12
> >
> > Thanks for the response. Some of my concerns have been answered. However, I think the current version of the evaluation using only one baseline is not ready for publication. Besides, I agree with the other reviewers that the discussion about original malicious functionality and payload modification is crucial. In fact, these points are also aligned with my comments on encrypted content and signature-based NIDS which fundamentally care about payload. When revising the paper, I suggest the authors include a better-described threat model to define your settings, such as where adversaries and NIDSes monitor the traffic and what their abilities are. For example, suppose they are all on the ISP side, given the majority of encrypted traffic nowadays. In that case, adversaries cannot modify payloads feasibly, nor can NIDSes use adequate signatures to detect malicious activities. For another, if the NIDSes are on the boundary of the end servers (which have the keys to decrypt the ciphertext), then things will be different. I would like to maintain my score.
> >
> > Again, I do like the authors considering the practicality of the generated traffic, as I've seen several works that do not have enough consideration while still getting published at high-level conferences and journals; maybe they "luckily" didn't meet expert-level reviewers as you did. This is why I encourage the authors to absorb the comments, in order to get ready for publication on top-tier venues like NeurIPS or Big 4 and make some real contributions to the community.

---

> > > ### Author Response · Authors · 2024-08-13
> > >
> > > Thank you for the detailed feedback and for highlighting key areas for improvement. We fully agree with the reviewer's suggestions and would like to provide an overview of the approach we will take in the revised paper to address the concerns.
> > >
> > > Regarding the baseline, we are currently conducting evaluation on the TANTRA model (Y. Sharon et.al, 2022), and will provide a valuable comparison to DiffuPac. The results of this evaluation will be included in the revised paper, offering a more comprehensive understanding of DiffuPac's performance relative to existing methods.
> > >
> > > We will also include a thoroughly detailed description of our threat model in the revised paper. This will clarify the settings in which DiffuPac operates, such as the positions of adversaries and NIDS, and their capabilities, especially in scenarios involving encrypted traffic and signature-based NIDS.
> > >
> > > We completely agree with the reviewer that many related works often overlook the evaluation of retaining malicious functionality or do not provide a fully detailed description of this crucial aspect. In our paper, we aim to set a new benchmark in the community by offering a comprehensive, fully-detailed account of how we conduct malicious functionality evaluations. Furthermore, we provide clear evidence that our model is capable of retaining malicious functionality, demonstrating a significant advancement in the field.

---

### Official Review · Reviewer_ogST · 2024-07-13

**Soundness:** 2
**Presentation:** 2
**Contribution:** 2
**Rating:** 4
**Confidence:** 3

**Summary:**

This work presents DiffPac, a method that leverages BERT and a diffusion model to generate adversarial packets aimed at evading network
intrusion detection systems. Compared to the approach by Han et al. (2021), DiffPac demonstrates superior effectiveness, with most machine learning-based classifiers trained on adversarial packets generated by DiffPac successfully evading detection across six different attack scenarios.

**Strengths:**

* Approach: The paper presents an interesting intuition for generating adversarial packets using a diffusion model, which is a creative approach in the field.

* Methodology: The methodology is clearly presented, making it easy to understand the steps and processes involved in the proposed DiffPac method.

**Weaknesses:**

* Motivation: The research motivation could be more clearly and concretely articulated. It would be beneficial to highlight the specific niche of this study in relation to existing work, such as Network Emulator (NetEM) and Metasploit (Homo-liak et al., 2018), Generative Adversarial Network (GAN) and Particle Swarm Optimization (PSO) (Han et al., 2021), and Reinforcement Learning (RL) (Hore et al., 2023). Rather than broadly stating that "their efficacy in evading detection in real-world conditions remains suboptimal" (line 64), the paper should detail the unique challenges and gaps that DiffPac addresses.

* Method for Preserving Packets’ Integrity: The approach to preserving packets' integrity by only mutating TCP flags, TTL, and window size, rather than the payload, is unrealistic for real-world conditions where payloads often contain the malicious behaviors. This limitation should be addressed to enhance the practical applicability of the method.

* Evaluation: The evaluation settings could be more closely aligned with those used by Han et al. (2021) to make it easier for the readers to understand the differences and contributions of this work.

**Questions:**

* Data Pre-processing: Could you explain the rationale and advantage of transforming IP numerical values into hexadecimal representation and treating them as discrete values for workpiece segmentation? What benefits does this approach offer?

* Formula (4): In Formula (4), does  S^{ben+mal}  denote z_t ?

* Dataset: How many adversarial packets are included in the test set? Providing a statistical description of the datasets used would enhance clarity and help understand the dataset composition.

**Limitations:**

The limitation section highlights three main concerns:

* The actual malicious functionality of the generated adversarial packets has not been thoroughly examined.

* There is inconsistent performance across the six attack scenarios tested.

* There is a potential risk associated with making the approach open-source, which could be exploited maliciously.

---

> ### Author Rebuttal · Authors · 2024-08-05
>
> We greatly appreciate the reviewers’ constructive review and insightful suggestions regarding our paper.
> Our responses are given in a point-by-point manner for each comment.
>
> **W1.**
>
> We agree that a more detailed discussion on the limitations of existing works and how DiffuPac addresses specific gaps will strengthen the motivation of our study.
>
> Traditional approaches, such as those by Homoliak et al., Han et al., and Hore et al., often rely on access to NIDS components or surrogate classifiers.
> These methods assume attackers have detailed knowledge of the NIDS, which is unrealistic in real-world scenarios.
> This dependency limits their practical applicability, as attackers typically operate with minimal insider knowledge of the target's system.
> DiffuPac addresses these limitations by adopting a classifier-free approach, leveraging pre-trained BERT and diffusion models to generate adversarial packets without relying on NIDS components or surrogate classifiers.
> This method allows DiffuPac to create adversarial traffic that mimics normal network behavior, making it more effective in evading detection in real-world conditions.
> Additionally, DiffuPac’s adaptive field alteration ensures that the modifications are contextually appropriate, maintaining the malicious functionality of the packets while enhancing their stealthiness.
> In the revised paper, we will expand on these points to highlight how DiffuPac innovates beyond existing methodologies, offering a practical solution for adversarial packet generation in real-world environments.
>
> **W2.**
>
> We understand the concern that in real-world conditions, payloads often contain the core malicious behaviors, and thus, not modifying the payload may seem limiting.
> Modifying the payload, which often contains core malicious behaviors, poses a significant risk of disrupting the packet's integrity, especially without detailed knowledge of its structure.
> We acknowledge the potential benefits of payload modification and plan to explore methods like Deep Packet Inspection (DPI) and selective payload adjustments in future work to enhance the realism of adversarial packets while maintaining their functionality.
>
> **W3.**
>
> We recognize the importance of such comparisons, particularly in demonstrating the unique contributions of DiffuPac in the context of existing methodologies.
> Our choice to diverge from the more complex evaluation settings used by Han et al. stems from the fundamentally different nature of our methodology.
> Since DiffuPac does not involve feature-level generation or direct interaction with NIDS components, the traditional metrics used in Traffic Manipulator's evaluation, which are deeply tied to feature extraction, are not directly applicable to our approach.
> Instead, by focusing on MER, we provide a clear and focused evaluation of DiffuPac's effectiveness in generating adversarial packets that evade detection in a more realistic context.
>
> We recognize the value of aligning more closely with other methodologies.
> In future work, we plan to expand our evaluation framework to include additional metrics and settings that facilitate direct comparisons with existing methods like Traffic Manipulator.
>
> **Q1.**
>
> We really appreciate the reviewer's inquiry.
> We convert IP numerical values to hexadecimal and treat them as discrete values for WordPiece tokenization to maintain the native structure and precision of network data.
> This approach ensures uniform tokenization across diverse protocol fields, preserves semantic meaning crucial for understanding network traffic patterns, and allows for efficient vocabulary management, enhancing the model's ability to accurately process and learn from the data.
>
> **Q2.**
>
> We sincerely apologize for any confusion caused by the insufficient explanation of Formula (4) in our original submission. We appreciate the reviewer's attention to detail and would like to clarify the notation and its context.
>
> To clarify, $ \mathbf{S}^\mathrm{ben \oplus mal} $ does not directly denote $ \mathbf{z}_t $ at each time step $ t $, where the original concatenated sequence $ \mathbf{S}^\mathrm{ben \oplus mal} $ is gradually perturbed.
>
> The expression $ q_\phi(\mathbf{z}_0 \mid \mathbf{S}^{\mathrm{ben \oplus mal}}) = \mathcal{N} (\text{EMB}(\mathbf{S}^{\mathrm{ben \oplus mal}}), \beta_0 \mathbf{I}) $ describes the initial state $ \mathbf{z}_0 $, which is derived from the embedding of the concatenated packet sequences.
>
> Subsequent steps in the diffusion process perturb this state to produce the series of latent variables $ \mathbf{z}_1, \mathbf{z}_2, \dots, \mathbf{z}_t $.
>
> We acknowledge that the initial explanation may not have fully conveyed these relationships, and we will ensure that the revision includes a more detailed and refined explanation.
>
> **Q3.**
>
> | Stage                             | Percentage of Total Data | Number of Normal Packets | Number of Malicious Packets |
> |-----------------------------------|--------------------------|--------------------------|-----------------------------|
> | Pre-training of the BERT model    | 60%                      | 5,079,021                | 2,734,858                    |
> | Fine-tuning phase                 | 20%                      | 1,693,007                | 911,619                      |
> | Training the classifier and NIDS  | 10%                      | 846,512                  | 455,819                      |
> | Testing phase                     | 10%                      | 846,504                  | 455,810                      |
>
> **Table 1**: Distribution of normal and malicious packets across different stages of data processing.
>
> In the testing phase, we included 455,810 malicious packets. DiffuPac generated an equal number of adversarial packets, ensuring a direct comparison. A detailed statistical description of the dataset composition, including normal and malicious packets across all stages of the process, will be included in the revised paper for enhanced clarity.

---

> > ### Comment · Reviewer_ogST · 2024-08-11
> >
> > Thank you for your response. I'm pleased to see the inclusion of payload modification and the plan to expand the evaluation settings in future work. I will be keeping my score unchanged.

---

### Official Review · Reviewer_CanA · 2024-07-13

**Soundness:** 3
**Presentation:** 2
**Contribution:** 2
**Rating:** 4
**Confidence:** 5

**Summary:**

This work proposes DiffuPac, an adversarial packet generation model that aims to evade Network Intrusion Detection Systems (NIDS) without depending on detailed knowledge of NIDS components. DiffuPac combines a pre-trained BERT model with a diffusion model. Experimental results show that DiffuPac outperforms traditional methods by 8.83%.

**Strengths:**

- This work studies an interesting and important issue about evading NIDS detection based on deep learning methods.
- I appreciate that the author divides the traffic data into mutable fields and immutable fields for modification.
- The experimental results support the effectiveness of the proposed method to evade detection.

**Weaknesses:**

- The paper achieves the following goals through expensive experiments (more than 30 hours of training on RTX 4090): select the normal packet sequence that is most similar to the malicious packet sequence, and modify the limited mutable fields (e.g. TCP flags, TTL, and window size) in the malicious packet sequence header according to the normal packet sequence. Given the very limited mutable fields modified within the packets and the lack of ablation studies in the experimental section, I doubt the necessity of using BERT and DiffuSeq. It seems that a simpler approach involving feature similarity matching, coupled with selective field selection and replacement, could potentially achieve the same goal without the complexity introduced by these advanced models.
- In the diffusion model where all mutable fields of the entire malicious packet sequence are regenerated, there is a concern about whether this process affects its original functionality. Although the modification effects on specific fields of individual packets are shown in the appendix, whether the author replays these generated packets in a real network environment to verify its functionality, rather than simply viewing them through Wirshark.
- Some minor issues: The paper needs improvement in writing quality. The connections between different components are not explicitly explained, and the specific format of the packet sequence input to the diffusion model is not detailed. In addition, the BERT-related content is very redundant in Sec. 3 and Figure 1. However, the related content is highly similar to ET-BERT and is only used to calculate the most relevant benign and malicious packet sequences in the "fine-tuning" stage (in fact, I think the operations in this stage do not belong to the category of "fine-tuning").

**Questions:**

Please see the above-mentioned weaknesses. I will confirm the author's response during the rebuttal.

**Limitations:**

The authors discussed the limitations in this work.

---

> ### Author Rebuttal · Authors · 2024-08-05
>
> We greatly appreciate the reviewers’ constructive review and insightful suggestions regarding our paper.
> Our responses are given in a point-by-point manner for each comment.
>
> **W1.**
>
> Simpler methods (e.g., [1], [2], [3]) often require surrogate classifiers or access to NIDS components.
> This is because the generated adversarial examples need to be evaluated and refined using these components to ensure they evade detection.
> The dependency on internal NIDS mechanisms limits the practicality of these methods in real-world scenarios where attackers typically lack such insider access.
>
> In contrast, DiffuPac operates independently of NIDS components by leveraging the capabilities of pre-trained BERT and diffusion models.
> This approach allows DiffuPac to intelligently alter specific packet fields, seamlessly blending malicious packets into normal traffic without needing detailed knowledge of the NIDS.
> The use of BERT is crucial in this process, as it provides the deep contextual understanding required to identify and modify fields in a way that maintains the packets' stealth and effectiveness.
>
> Our ablation study compared DiffuPac's performance with a standard transformer trained from scratch against the BERT-based model.
> The results clearly demonstrate the BERT-based model's superiority in both evasion capabilities and accurate packet reconstruction. This highlights the essential role of BERT in DiffuPac, enabling the model to make sophisticated, contextually-aware modifications that simpler approaches, which lack such depth of understanding, cannot achieve.
> We kindly ask the reviewer to refer to the "Ablation Study" section in the main author rebuttal for more detailed insights into the necessity of BERT in DiffuPac's architecture.
>
> We agree with the reviewer that simpler approaches could be effective when contextual relevance with normal packets is identified.
> This will be an important topic for future research, where we will explore the potential of such methods in conjunction with or as alternatives to more complex models like BERT and diffusion models.
>
> **W2.**
>
> We really appreciate the reviewer's concern regarding the capability of DiffuPac in retaining the malicious functionality of generated adversarial packets.
> To address this concern, we conducted a detailed malicious functionality evaluation within a controlled, isolated network environment.
> We kindly ask the reviewer to refer to the "Malicious Functionality Evaluation" section in the main author rebuttal for a more detailed explanation.
> Currently, we are also conducting malicious functionality evaluation on Brute Force Attacks, and other type attacks. These experimental results and their fully-detailed explanations will be added in the revised paper.
>
> **W3.**
>
> We sincerely appreciate the reviewer’s insightful feedback and understand the importance of refining our explanations.
> Our methodology involves a structured process where each component plays a crucial role:
>
> - **Pre-Training Phase:**
>   The BERT model is pre-trained using tasks like the Masked Unidirectional Flow Model and Same Sequence-origin Prediction. These tasks are essential for BERT to develop a deep understanding of network traffic patterns and dependencies, which are critical for the later stages.
>
> - **Fine-Tuning with Diffusion Models:**
>   In this phase, the pre-trained BERT model enhances the diffusion process, allowing for the generation of adversarial packets that are contextually aligned with normal traffic. The packet sequences are formatted with token, positional, segment, and time step embeddings, providing a comprehensive input structure that the diffusion model uses to accurately reconstruct and generate packets.
>
> In the revised paper, we will take the following steps to improve clarity and conciseness:
>
> - **Enhanced Component Integration:**
>   We will provide a more explicit explanation of how each component of our methodology connects and contributes to the overall process. This will include clearer descriptions of the role of embeddings and how they prepare the data for effective processing by the diffusion model.
>
> - **Streamlining BERT-Related Content:**
>   To address concerns about redundancy, we will condense the BERT-related sections, focusing on its unique application in DiffuPac. This will involve removing repetitive content and emphasizing only the most critical aspects that highlight DiffuPac’s innovations.
>
> - **Revisiting Terminology:**
>   We will re-evaluate and revise terminology, particularly around the "fine-tuning" stage, to ensure it accurately reflects the specific operations and the unique way BERT is used in DiffuPac. This will help avoid any confusion and better align with the expectations of readers familiar with existing methodologies.
>
> - **Detailed Input Format Explanation:**
>   We will include a more detailed explanation of the packet sequence input format to the diffusion model, ensuring readers understand the significance of each embedding type and how they contribute to the model’s performance.
>
> [1] Homoliak et al., Improving Network Intrusion Detection Classifiers by Non-Payload-Based Exploit-Independent Obfuscations
>
> [2] Hashemi et al., Towards Evaluation of NIDSs in Adversarial Setting
>
> [3] Kuppa et al., Black Box Attacks on Deep Anomaly Detectors

---

> > ### Comment · Reviewer_CanA · 2024-08-10
> >
> > Thanks to the authors for their response. I have reviewed the rebuttals. However, showing only the case of Port Scan attack cannot solve my concern about whether DiffuPac affects the original functionality of the traffic. Once the original functionality is destroyed, the practical value of this work becomes questionable. Therefore, this should be a primary consideration before designing the method. In addition, I suggest that the authors carefully check their experimental results, as the BERT-based results in the rebuttal PDF (Table 1) seem inconsistent with those in the submitted manuscript (Table 2). Overall, I am inclined to keep the original scores.

---

> > > ### Author Response · Authors · 2024-08-13
> > >
> > > We deeply appreciate the response from the reviewer.
> > >
> > > First of all, we sincerely apologize for not fully addressing the concerns during the rebuttal phase. Due to time and resource constraints, we were only able to include the Port Scan attack evaluation. However, we have completed the evaluation of Brute Force attacks and are pleased to report that the results are promising. The responses from both the original Brute Force packets and the adversarial Brute Force packets indicate successful SSH access, demonstrating that DiffuPac preserves the original functionality of the traffic. We will include these results in the revised paper, along with evaluations of other attack types. We hope these additional evaluations will alleviate the reviewer's concerns.
> > >
> > > Regarding the inconsistency between the BERT-based results in the rebuttal PDF (Table 1) and the submitted manuscript (Table 2), we deeply appreciate the feedback and acknowledge the discrepancy. We suspect that the variations might be due to potential minor differences in the random seeds used during training.
> > >
> > > Despite these minor differences, the overall trends remain consistent, and the results continue to demonstrate the promising capabilities of our model. Moving forward, we will do our utmost to ensure that the revised paper reflects consistent and verified results. We will carefully re-run experiments, double-check our settings, and align all reported data to maintain accuracy and clarity.

---

### Author Rebuttal · Authors · 2024-08-05

We sincerely appreciate the reviewer’s thoughtful feedback and constructive insights.
Here, we summarized the experiments that we conducted for the ablation study and maliciousness functionality evaluation.
These experiments will be included in the revised version of the paper.

### Ablation Study

**Objective:** This ablation study assesses the impact of substituting the pre-trained BERT model with a standard transformer architecture in the denoising process of DiffuPac.
The focus is on assessing each model's ability to accurately reconstruct and generate adversarial packets that can effectively evade detection.

**Experimental Setup:**

- **Non-BERT-based:** This variant of DiffuPac employs a standard transformer architecture for the denoising process, as outlined in the DiffuSeq methodology [1].
    Notably, this model is trained from scratch without leveraging any pre-trained weights, thus relying solely on the information provided during the training phase.
- **BERT-based:** The existing DiffuPac configuration, which incorporates a pre-trained BERT model for the denoising process, was utilized.
    This setup takes advantage of BERT's pre-trained weights, enabling enhanced semantic understanding and contextual relevance during packet reconstruction.

**Training and Evaluation:** Both models were trained under identical conditions and evaluated based on their success in generating adversarial packets that evade detection.
The results are shown in the accompanying PDF.

**Result And Analysis:** As shown in Table 1, the BERT-based DiffuPac significantly outperforms the non-BERT variant, demonstrating superior ability to reconstruct packets that blend seamlessly into normal traffic.

The BERT-based model excels in pairing malicious packets with normal packets that exhibit high contextual relevance. This strategic pairing allows the generated adversarial packets to blend seamlessly into normal traffic, thereby increasing their likelihood of evading detection.
In contrast, the non-BERT-based model, which lacks this pre-trained knowledge, resorts to random matching of malicious and normal packets, leading to a noticeable reduction in the effectiveness of the generated adversarial traffic.

An intriguing aspect of DiffuPac is its dual utilization of the pre-trained BERT model: first, as the denoising engine during the packet reconstruction process, and second, as a critical tool for identifying contextually relevant packet pairs.
To the best of our knowledge, this dual application of a pre-trained model within a diffusion framework is unprecedented, further highlighting the innovation and effectiveness of our approach.


### Malicious Functionality Evaluation

**Objective:** To validate that DiffuPac-generated adversarial packets retain their original malicious functionality.

**Experimental Setup:** We evaluated DiffuPac’s efficacy in generating adversarial packets that maintain their malicious functionality in a controlled test environment using UTM hypervisor technology.
This setup included two virtual machines: one running Kali Linux (attacker) equipped with Nmap, and another running Ubuntu 24.04 (victim). The isolation ensured a realistic yet controlled network environment. The results are shown in the accompanying PDF.


**Initial Attack Demonstration:**
Before proceeding to evaluate the generated adversarial packets of DiffuPac, we first demonstrated a successful Port Scan attack within our controlled environment.
This initial step is a fundamental step to ensure the validity and reliability of subsequent tests with both original and adversarial packets.
The network configurations depicted in Figures 1(a) and 1(b) confirm that both virtual machines were correctly placed within the same subnet.
As demonstrated in Figure 1(c), the Port Scan successfully identified open ports on the Ubuntu, highlighting port 22 as vulnerable.

**Detailed Methodology:**

- **Initial Packet Capture:** We initiated a Port Scan attack from the Kali Linux machine targeting the Ubuntu server.
    During the execution of this attack, we captured all packets sent from the attacker to the victim, saving them as a pcap file.
    This capture was timed precisely to begin as the Nmap tool launched the Port Scan, ensuring that only the relevant src-to-dst packets were recorded.
    This focus aligns with the operational framework of DiffuPac, which concentrates on modifying packets sent from the source.
- **Adversarial Packet Generation and Replay:** Using the captured src-to-dst packets, we employed DiffuPac to generate adversarial Port Scan packets, which were then saved in a separate pcap file.
    These adversarial packets were subsequently replayed using the Tcpreplay tool, directing them towards the Ubuntu server.
    Concurrently, we used Wireshark on the Ubuntu machine to capture the incoming responses, allowing us to analyze the effectiveness of the adversarial packets in real-time.
- **Fair Comparative Analysis:** To ensure a balanced evaluation, we also replayed the original Port Scan pcap file using Tcpreplay, capturing the responses in Wireshark on the Ubuntu side. This dual replay enabled a direct comparison between the packets' responses generated by the original and adversarial Port Scan packets.

**Results And Analysis:**
As depicted in Figures 2(a) and 2(b), the response to the adversarial Port Scan packets closely mirrors that of the original Port Scan.
A detailed inspection of the TCP flags reveals that packets with SYN and ACK flags targeted port 22, corroborating the initial vulnerability identified in Figure 1(c).
This consistent response across both original and adversarial packets provides compelling evidence that DiffuPac successfully generates adversarial packets that retain their intended malicious functionality.

Currently, we are also conducting evaluation on other type of attacks, that will be included in the revised paper.

[1] Gong S et. al., Diffuseq

---

> ### Comment · Area_Chair_8iM6 · 2024-08-10
>
> Hi Authors,
>
> I have noticed that you used the CICIDS2017 dataset to carry out your experiments. Were you aware that the CICIDS17 dataset is flawed?
>
> This fact has been known since 2021 in the ML-based NIDS community. See: [Engelen, Gints, Vera Rimmer, and Wouter Joosen. "Troubleshooting an intrusion detection dataset: the CICIDS2017 case study." 2021 IEEE Security and Privacy Workshops (SPW). IEEE, 2021.].
>
> Do note that even its extended version (CICIDS2018) is also flawed; this was found in the 2022 best paper award at CNS (see [Liu, Lisa, et al. "Error prevalence in nids datasets: A case study on cic-ids-2017 and cse-cic-ids-2018." 2022 IEEE Conference on Communications and Network Security (CNS). IEEE, 2022.])

---

> > ### Author Response · Authors · 2024-08-13
> >
> > Thank you for bringing this important issue to our attention. We deeply appreciate your feedback regarding the CICIDS2017 dataset and its known flaws, as highlighted in the works by Engelen et al. (2021) and Liu et al. (2022).
> >
> > In the field of adversarial attack generation for cybersecurity, CICIDS2017 and its extended version, CICIDS2018, are indeed widely used and recognized datasets. Prominent studies, including recent research like Deep Pack Gen (supported by the U.S. Military Academy) and TANTRA (2022), have continued to utilize these datasets. Additionally, influential surveys, such as "Adversarial Machine Learning for Network Intrusion Detection Systems: A Comprehensive Survey," acknowledge CICIDS2017/2018 as standard benchmarks in the field.
> >
> > Given the pioneering nature of our work in generating adversarial packets, we aimed to align with the commonly accepted practices in the community by using these datasets. However, we were unaware of the specific flaws that have been mentioned, as they have not been widely addressed in the literature we have encountered. We genuinely appreciate the guidance on this matter.
> >
> > In response to this valuable feedback, we have already conducted experimental evaluations using a refined version of the CICIDS2017 dataset. The preliminary results from these evaluations indicate an even higher evasion rate than initially reported, which is a promising outcome. We are currently extending these evaluations to the refined version of the CICIDS2018 dataset and will include the results in our revised paper.
> >
> > Furthermore, in our revised paper, we will include a thorough discussion on the necessity of using the refined CICIDS2017/2018 datasets for evaluating adversarial attack generation. We believe this discussion will have a significant impact on the community, as it addresses a critical issue that has often been overlooked. By introducing this dialogue, we aim to set a new benchmark for the field.
> >
> > Given that our work introduces a first-of-its-kind adversarial packet generation model, we are confident that this discussion will draw substantial attention from the community. We hope to inspire further research that prioritizes dataset integrity, ultimately contributing to the ongoing improvement of dataset quality in the ML-based/DL-based NIDS community.

---

### Decision · Program_Chairs · 2024-09-25

**Decision:**

Accept (poster)

**Comment:**

The paper provides a novel method to evade Machine Learning-based Network Intrusion Detection Systems (ML-NIDS). From a security viewpoint, the major contribution of the paper is that the proposed method can work even in a very constrained setting wherein the attacker has no knowledge/access to the targeted ML-NIDS. Such a property is provided by leveraging models "pre-trained" on (well-known) datasets. The experiments validate the effectiveness of the method.

No reviewer pointed out critical flaws in the evaluation, and all reviewers praised the quality of writing of the paper. The major concerns on this paper were related to the guarantee that the method achieves evasion while preserving the original malicious functionality of the generated data. The authors have responded to this concern by carrying out some validatory experiments during the rebuttal. Nonetheless, such a verification is challenging to carry out, and few works have addressed this aspect (which is crucial from a security viewpoint).

Given the above, the paper does provide a meaningful contribution in the domain of "applied" ML for cybersecurity, and while the results are not groudbreaking and may present some uncertainties, there is reason to believe that this paper will inspire future work in this domain.

I also favor making the discussion between authors and reviewer publicly visible. There is much to learn in the comments/remarks generated during the discussion phase.